# Constraints-of-Thought: A Framework for Constrained Reasoning in Language-Model-Guided Search

## Abstract

While researchers have made significant progress in enabling large language models (LLMs) to perform multi-step planning, LLMs struggle to ensure that those plans align with high-level user intent and satisfy symbolic constraints, especially in complex, multi-step domains. Existing reasoning approaches such as Chain-of-Thought (CoT), Tree-of-Thought (ToT), and verifier-augmented methods, expand the search space but often yield infeasible actions or hallucinated steps. To overcome these limitations, we propose *Constraints-of-Thought* (Const-o-T), a framework that provides a structured prior that enables Monte Carlo Tree Search (MCTS) focus search on semantically meaningful paths. Each reasoning step is represented as an ⟨intent, constraint⟩ pair, which serves both to compress the search space and enforce validity. Unlike prior methods that merely generate reasoning traces or validate outputs post hoc, Const-o-T uses ⟨intent, constraint⟩ pairs to actively focus the search toward feasible and meaningful plans. We integrate Const-o-T into MCTS using a structured representation of intent–constraint pairs: constraints prune infeasible branches and guide exploration toward semantically valid actions, improving planning efficiency and verifiable decision-making. We demonstrate across three domains: Risk game, CAD code generation , and arithmetic reasoning that our approach outperforms baselines, yielding higher accuracy and stronger structural alignment. Our contribution is to demonstrate that Const-of-T offers a generalizable foundation for constraint-guided reasoning, enabling more efficient, constraint-aligned, and domain-adaptable planning with LLMs.

## 1 INTRODUCTION

Planning is a fundamental challenge in AI, particularly in domains requiring natural language understanding and complex, multi-step reasoning (Russell et al., 1995; Ghallab et al., 2004). LLMs, such as GPT (OpenAI, 2023) and LLaMA Touvron et al. (2023), have demonstrated impressive capabilities in generating plans from textual descriptions, yet they struggle to ensure these plans align with user intent and satisfy domain-specific constraints (Zhou et al., 2024; Turpin et al., 2023).

LLM-based approaches to planning, such as those that leverage Chain-of-Thought (CoT) reasoning Wei et al. (2022), appear to "think through" the space of possible plans. However, in practice, these approaches frequently hallucinate infeasible steps. These weaknesses are exacerbated in NP-hard domains, e.g. strategy games Silver et al. (2017); Guan et al. (2024) and program synthesis Madaan et al. (2022). Even in polynomial-time domains like multi-step arithmetic reasoning Cobbe et al. (2021), strict validity constraints present key difficulties for LLM-based planning.

Consider this example arithmetic reasoning problem: *"A factory packs 12 pencils into each box and 18 boxes into each carton. If the factory produces 10 cartons, and 240 pencils are found defective, how many non-defective pencils remain?"* A typical CoT reasoning might generate: *'10 cartons × 12 pencils = 120 pencils; subtract 240 defective = -120 pencils."* Without any mechanisms to validate the intermediate reasoning-steps, the reasoning-trace ignores critical problem constraints. For example, multiplying by the number of boxes per carton (18) is missing. As a result, the reasoning produces an answer that is not only numerically incorrect but also semantically invalid. CoT aims to reason step by step, but without mechanisms for grounding or constraint enforcement, it

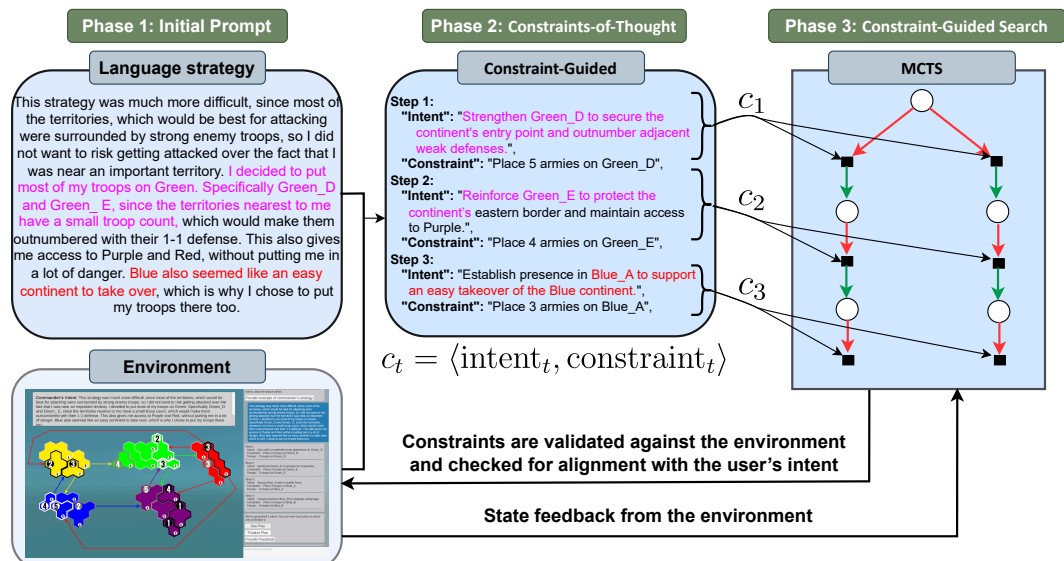

Figure 1: Const-of-T empowers LLMs to (i) infer intent statements, and (ii) extract a corresponding constraint from a high-level strategy, guiding MCTS toward optimal, rule-compliant actions.

often generates solutions that contradict the problem structure or violate basic logical consistency. Instead, a good solution should produce grounded, constraint-aware reasoning that aligns with both the problem state and the high-level task description.

We propose Constraints-of-Thought (Const-o-T), a reasoning framework that translates unconstrained natural language strategies into structured, verifiable constraints. Rather than producing free-form reasoning traces, Const-o-T decomposes each step into: (i) an *intent* statement describing the high-level strategic goal, and (ii) a corresponding *constraint* that can be symbolically verified. This structured representation bridges user guidance with downstream symbolic verification.

While Const-o-T is useful in isolation, we demonstrate its full potential by integrating it with Monte Carlo Tree Search (MCTS) for constraint-guided planning. In this combined system, an LLM first produces a sequence of ⟨intent, constraint⟩ pairs based on the user's strategic input. These constraints then act as symbolic controllers during MCTS rollouts: pruning infeasible actions, reducing the branching factor, and focusing search on semantically valid paths. This modular integration enables more efficient planning and tighter alignment with user intent. Const-o-T was evaluated across three domains with distinct challenges where it consistently outperformed baselines. These results demonstrate that structured intermediate representations help align outputs with user intent, reduce search space, and enable symbolic verification. Our work presents four key contributions:

1. **Constraints-of-Thought (Const-o-T).** We propose a novel reasoning framework that decomposes high-level strategies into paired ⟨intent, constraint⟩ representations, enabling symbolic verification and alignment with user intent.

2. **Constraint-Guided MCTS.** We integrate Const-o-T with MCTS, pruning infeasible branches and steering search toward semantically meaningful actions, which improves planning efficiency and goal alignment.

3. **Computational Evaluation.** We validate Const-o-T across three domains (§ 3), demonstrating improved performance under diverse and challenging domains.

4. **Human Evaluation.** Through a user study on the board game Risk, we show that our approach supports planning with reasoning that is more transparent, aligned, and usable than alternative methods ($p < 0.05$).

## 2 METHOD: CONSTRAINTS-OF-THOUGHT FRAMEWORK GUIDED SEARCH

We address task planning in complex domains where translating natural language strategies into executable plans remains challenging. Directly mapping user instructions to actions often results in errors, ambiguity, and infeasible exploration. We introduce Const-o-T, a structured reasoning

framework that guides MCTS toward semantically meaningful and constraint-satisfying paths. We address task planning in complex domains where translating natural language strategies into executable plans remains challenging. Directly mapping user instructions to actions often results in errors, ambiguity, and exploration of infeasible paths. We introduce Const-o-T, a structured reasoning framework that guides MCTS toward semantically meaningful and constraint-satisfying paths.

## 2.1 TASK PLANNING

We frame task planning as a Partially Observable Markov Decision Process (POMDP) $(\mathcal{S}, \mathcal{A}, T, r, \Omega, \gamma, H)$, where the agent must generate a finite-horizon action sequence based on partial observations and domain knowledge. At each step, the agent selects a sequence of actions $(a_0, \ldots, a_{H-1})$ to maximize a reward function. The state space, $\mathcal{S}$, captures all relevant information about the environment . The action space, $\mathcal{A}$, contains all actions available to the agent in a given state. A history trajectory, $h_t$, is defined as the sequence of observations and actions up to time $t-1$, i.e., $h_t = (o_0, a_0, o_1, a_1, \ldots, o_{t-1}, a_{t-1})$.

The transition dynamics, $T$, determine how states evolve in response to actions, varying by domain, i.e. they may be stochastic or deterministic depending on the dynamics of the domain. The evaluation function or reward, $r$, serves as an evaluation measure of the generated output's utility, which can be derived from a state-based fitness function or from external evaluators.. In contrast, for CAD and arithmetic domains, $r$, is approximated using LLM evaluations of code validity or solution correctness. The discount factor, $\gamma$, balances immediate and future rewards.

Rather than solving for an optimal policy $\pi^*$, we approach planning as an online search problem, where the agent aims to construct an action sequence $(a_0, \ldots, a_{H-1})$ that maximizes cumulative reward given by Eq. 1-3, where, $\mathbf{1}\{\cdot\}$, is an indicator function that equals 1 if the generated action satisfies the constraints and 0 otherwise, and, $F(s, a)$, is a task-specific evaluation function.

$$R_{\text{total}} = \sum_{t=0}^{H-1} \gamma^t r(s_t, a_t) \tag{1}$$

$$r(s, a) = \mathbf{1}\{\text{constraint satisfied}\} + F(s, a) \tag{2}$$

$$F(s, a) = \begin{cases} z_1(s, a), & \text{Risk (fitness function, see § A.6)} \\ z_2(s, a), & \text{CAD code (LLM-as-a-Judge)} \\ z_3(s, a), & \text{Math arithmetic (LLM-as-a-Judge)} \end{cases} \tag{3}$$

## 2.2 CONSTRAINTS-OF-THOUGHT

We introduce *Constraints-of-Thought* (Const-o-T), a framework that represents structured reasoning steps guiding the agent's decision-making process. Const-o-T encodes each reasoning step as a pair: a high-level strategic *intent* and a corresponding formal *constraint*. Let the environment state at time $t$ be $s_t \in \mathcal{S}$, and let the available action space be $\mathcal{A}(s_t)$. A Const-o-T step is defined as a tuple, $c_t = \langle \text{Intent } i_t, \text{Constraint } c_t \rangle$, where **Intent** $i_t \in \mathcal{I}$: A natural language description of the strategic reasoning (e.g., "Reinforce a border to deter enemy") and **Constraint** $c_t \in \mathcal{C}$: A machine-executable symbolic instruction (e.g., "Place 5 troops on Territory A") Once extracted, these $\langle \text{Intent } i_t, \text{Constraint } c_t \rangle$ pairs serve a dual role in planning. The intent provides a human-interpretable explanation of the agent's reasoning, while the constraint restricts the feasible action space by pruning actions that violate domain rules. Formally, given a state $s_t$, the constraint defines a reduced action set, $A'(s_t) \subseteq A(s_t)$, containing only actions consistent with Constraint$_t$. This structured representation enables the agent to generate plans that are both aligned with the user's high-level strategy and executable within the environment.

## 2.3 CONSTRAINT-GUIDED MONTE CARLO TREE SEARCH

We integrate Const-o-T with MCTS to generate plans that align with user strategic intent. Standard MCTS often fails to reflect user preferences in complex environments due to vast, unstructured search spaces. Our constraint-guided approach addresses this limitation by using extracted constraints to direct each phase of the MCTS process.

**Algorithm Overview:** Given a natural language strategy, we first extract a sequence of $\langle$intent, constraint$\rangle$ pairs using a language model. These extracted constraints subsequently guide the MCTS

exploration process to generate strategy-aligned actions. The complete algorithmic details are provided in Appendix (§A.2).

**Selection Phase:** Starting from the root, MCTS recursively selects child nodes using a modified Upper Confidence Bound (UCB) criterion that balances exploration and exploitation with the LLM's confidence, as given by Eq. 4, where $\lambda$ represents the log-probability weight from the LLM's constraint-guided action prediction.

$$\text{UCB}(v, a) = Q(v, a) + c_{\text{uct}} \sqrt{\frac{\ln(1 + N(v))}{1 + N(v, a)}} + \lambda \cdot \log P_{\text{LM}}(a \mid \text{state}, \text{constraint}) \qquad (4)$$

**Constraint-Guided Search.** Our key innovation uses the total number of extracted constraints to determine the MCTS rollout budget, ensuring that search effort scales with strategy complexity. Each rollout is guided by one constraint, restricting exploration to paths consistent with user intent.

**Expansion and Evaluation.** At leaf nodes, LLMs generate candidate actions given the current state, active constraint, and user strategy. We filter candidates for legality (e.g., game rules) and constraint satisfaction before expansion. Each resulting child node is evaluated using a domain-specific fitness function measuring strategic objective achievement and constraint adherence.

## 3 TASKS

We evaluate our proposed Const-o-T framework across three diverse domains that present distinct yet complementary challenges for structured reasoning and planning: (i) strategic gameplay (Risk), where players must translate natural language strategies into troop placements on a combinatorially rich territory map; (ii) CAD code generation, which requires grounding natural language descriptions into executable parametric code that yield valid 3D models; and (iii) arithmetic reasoning, which demands step-by-step symbolic manipulation to perform numerical operations. These domains offer a comprehensive validation of the utility of Const-o-T towards improving performance and correctness on tasks demanding topological, geometric, and symbolic reasoning..

### 3.1 GAME STRATEGY: RISK GAME

**Task Description.** Risk is a strategy board game played on a world map of 21 territories grouped into six continents. While the full game involves sequential phases of reinforcement, attack, and fortification across many turns, our focus is restricted to the very first phase of the first turn: the initial troop placement studied in prior work (Tambwekar et al., 2023; Dodeja et al., 2024) (see Appendix §A.8 for illustrative example).

In this setting, each player begins with an equal number of troops and must allocate them to unoccupied territories. The strategic challenge lies in distributing troops across the map to balance continental control, defensive positioning, and flexibility for future expansion. Because no battles or fortifications occur during this initial phase, the task reduces to predicting players' placement decisions given their stated natural language strategies. This controlled setting offers a clear window into how LLMs handle constraint-guided strategic decision-making (Guan et al., 2024; Xu et al., 2025; , FAIR), providing insights into their broader reasoning abilities beyond text generation.

**Dataset.** We use the Commander's Intent (CI) dataset introduced by (Tambwekar et al., 2023), which contains $1,053$ examples of natural language strategies paired with corresponding ground-truth territories for troop placement. The dataset provides no ground-truth annotations for attack, reinforcement, or free movement phases. Therefore, we evaluate these additional phases separately through a user study.

### 3.2 CAD CODE GENERATION

**Task Description.** CAD code generation involves translating natural language descriptions of 3D objects into parametric Computer-Aided Design (CAD) scripts (e.g., using CADQuery), which can be directly compiled into executable 3D models. The task is challenging because generated code must ensure both syntactic correctness and semantic fidelity, accurately reflecting the intended geometric and structural properties. These dual demands make CAD code generation a compelling

testbed for reasoning frameworks like Const-o-T, where explicit constraints can help models produce outputs that are not only valid but also verifiably aligned with user intent.

**Dataset.** We evaluate LLMs using the CADPrompt benchmark Alrashedy et al. (2025), a dataset containing 200 3D design examples, each paired with a natural language description written by a novice human and a corresponding Python script written by a CAD design expert (Figure 10).

### 3.3 ARITHMETIC REASONING

**Task Description.** Arithmetic reasoning requires solving multi-step mathematical word problems by correctly mapping natural language descriptions into symbolic equations and step-by-step calculations. This makes the domain a strong testbed for evaluating whether reasoning frameworks can generate logically consistent intermediate steps that lead to the correct numerical answer. For LLMs, the challenge lies in grounding natural language into the correct mathematical formulation, maintaining coherence across steps, and avoiding spurious reasoning paths.

**Dataset.** We evaluate LLMs using the GSM8K benchmark (Cobbe et al., 2021), which contains 8.5K high-quality math word problems, split into approximately 7.5K training and 1K test instances. Following standard zero-shot evaluation protocol, we use only the test set in our experiments to assess the performance of our proposed Const-o-T approach. GSM8K is widely regarded as a robust dataset for benchmarking multi-step math problem solving in LLMs.

## 4 EXPERIMENTS

Our experiments are designed to evaluate the following key hypotheses: **(1) Accuracy and Alignment:** Const-o-T yields higher accuracy and stronger structural alignment with user intent than unconstrained baselines (Direct Prompt, CoT, ToT, or vanilla MCTS). **(2) Hallucination Reduction:** Constraint-guided reasoning decreases invalid or over-generated outputs, producing plans that more closely match ground truth in length and feasibility. **(3) Human Understanding:** The use of ⟨intent, constraint⟩ pairs provides clearer and more aligned intermediate reasoning steps than raw CoT traces, supporting human understanding and evaluation across domains.

### 4.1 GAME STRATEGY: RISK GAME

**Baselines.** We compare Const-o-T against a diverse set of baselines capturing both prompting-only and search-integrated methods. (1) *Direct Prompt* provides answers without a reasoning structure, while CoT introduces free-form step-by-step reasoning but lacks grounding in constraints. (2) *ToT* extends CoT by exploring multiple reasoning paths, and (3) *CoT + Rejection Sampling (RS)* filters out inconsistent outputs post hoc. We also include LLMFP (Hao et al., 2025), a recent framework that translates natural language problems into formal optimization constraints and solves them with external solvers. From the search perspective, we evaluate standard MCTS without reasoning guidance and MCTS with CoT, which leverages free-form reasoning within search. As an additional baseline, we used classical constraint optimization guided by an LLM to supply active constraints; see Appendix § A.9 for details.

**Metrics.** For troop placement, we measure accuracy (proportion of correctly predicted placements) and F1-score (balancing precision and recall for territory prediction quality). Additional game phases (e.g. the attack, reinforcement, and free movement) are evaluated qualitatively through expert user studies, as no ground truth annotations exist for these tasks.

**Results.** The results in Table 1 show that constraint-guided reasoning provides consistent gains over unconstrained baselines. Direct prompting and CoT achieve reasonable performance but frequently lack specificity and structural validity, while CoT + RS and ToT offer only marginal improvements. We also include LLMFP which achieves competitive accuracy on GPT-4 (84%), but its F1-scores remain substantially lower across all models. This gap highlighted the lack of alignment with territory-level predictions. On the search side, MCTS with Const-o-T delivers the best overall performance, surpassing both vanilla MCTS and MCTS with CoT, achieving up to 86% accuracy and 0.78 F1-score. These results confirm that explicitly grounding reasoning in constraints not only reduces invalid outputs but also guides search toward plans that more reliably align with user intent.

Table 1: Accuracy and F1-score for strategic game (Risk) troop placement plans.

| Method | LLaMA-3.3-70B | | GPT-4 | | DeepSeeK-R1 | | GPT-OSS-120B | | GPT-OSS-20B | |
|---|---|---|---|---|---|---|---|---|---|---|
| | Acc. | F1 | Acc. | F1 | Acc. | F1 | Acc. | F1 | Acc. | F1 |
| Direct Prompt | 78% | **0.74** | 79% | 0.75 | 70% | 0.68 | 79% | **0.75** | 77% | 0.73 |
| CoT | 74% | 0.69 | 81% | **0.79** | 67% | 0.64 | 77% | 0.74 | 78% | 0.73 |
| CoT + RS | 78% | 0.73 | 77% | 0.77 | 68% | **0.69** | 79% | **0.75** | 76% | 0.72 |
| LLMFP | 78% | 0.59 | **84%** | 0.70 | 69% | 0.56 | 63% | 0.53 | 74% | 0.62 |
| Const-o-T | **81%** | 0.72 | 83% | 0.76 | **78%** | 0.67 | **86%** | **0.75** | **81%** | **0.73** |
| ToT | 79% | 0.59 | 83% | 0.70 | 80% | 0.62 | 78% | 0.62 | **82%** | 0.61 |
| MCTS | 80% | 0.67 | 84% | 0.71 | 79% | 0.61 | 82% | 0.65 | **82%** | 0.64 |
| MCTS with CoT | 81% | 0.65 | 84% | 0.70 | **82%** | 0.59 | 79% | 0.67 | **82%** | 0.64 |
| MCTS with Const-o-T | **83%** | **0.72** | **86%** | **0.78** | 77% | **0.70** | 85% | **0.76** | 81% | **0.71** |

**Error Analysis.** To better understand the qualitative behavior of the search strategies, we analyze the distribution of generated plan lengths relative to the ground truth. Figure 2 presents the percentage of outputs that are shorter, equal in length, or longer than the ground truth across the three methods. Standard MCTS and MCTS with CoT show a clear tendency to over-generate, with more than 55% of plans exceeding the GT length. In contrast, MCTS with Const-o-T substantially reduces this bias: nearly half of the generated plans, 46.8%, match the ground truth length, while the proportion of longer plans decreases to 44.2%. This shift indicates that constraint-guided search not only improves structural alignment but also mitigates length overestimation, producing outputs that are more consistent with the average length of the target outputs.

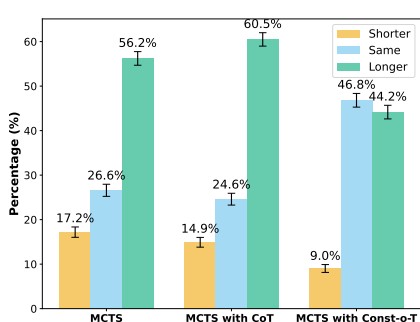

Figure 2: Distribution of plan lengths relative to ground truth for GPT-4.

**User study.** We conducted a within-subjects user study with $n = 18$ participants under an Internal Review Board (IRB) approved protocol to test whether our Constraint-Guided MCTS can support users in playing the boardgame, Risk. Users provide unstructured natural language describing their "commander's intent," and our method would synthesize a corresponding plan. Unlike the computational results in Table 1, which focus only on the initial troop placement phase, this user study evaluates AI support across all three game phases (i.e., Reinforce, Attack, and Freemove) over multiple complete turns of the game.

Our experiment had one independent variable, *interaction mode*, with three levels: (1) *Aligned*, where the system follows the user's instructions; (2) *Agnostic*, where it seeks only to win; and (3) *Adversarial*, where it intentionally acts against the user's intent. This adversarial condition was included to address the potential confound inherent in adaptive approaches . We measured four dependent variables using 7-point Likert scales: Transparency Silva et al. (2023), Usability Lewis (2018), Trust, and Alignment. Details of the study and scales used are provided in Appendix § A.4.

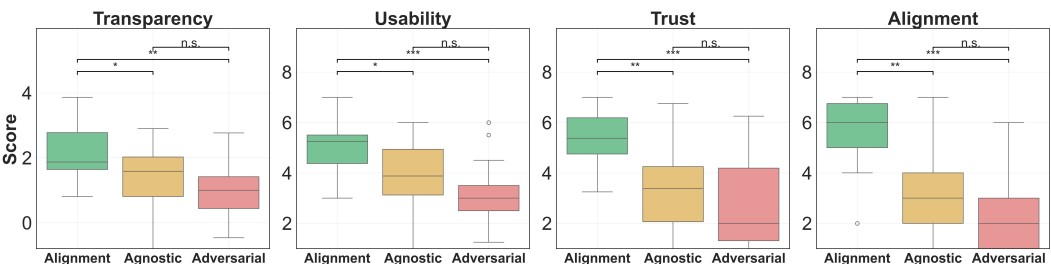

Figure 3: User study ratings across three interaction modes: alignment, agnostic, and adversarial. Statistical significance is indicated by asterisks ($^*p < 0.05$, $^{**}p < 0.01$, $^{***}p < 0.001$).

We performed a one-way ANOVA for transparency and usability and the Kruskal-Wallis test for trust and alignment. We found statistically significant differences in transparency ($F(2, 51) = 7.78$, $p < 0.01$), usability ($F(2, 51) = 10.23$, $p < 0.001$), trust ($H(2) = 19.80$, $p < 0.001$), and alignment ($H(2) = 23.75$, $p < 0.001$) across interaction modes, as shown in Fig. 3. Pair-wise comparisons using a Tukey's HSD test and Dunn's test showed that the aligned interaction mode (i.e., our approach) was viewed statistically significantly better than the agnostic and adversarial.

These results demonstrate statistically significantly that our approach was able to better support users in playing the boardgame Risk than an LLM-based decision support system (DSS) that merely ignored the user and sought to win (i.e., agnostic) or an LLM-based DSS that was still adaptive but did not capture user intent (i.e., adversarial).

## 4.2 CAD CODE GENERATION

**Baseline.** We evaluate CoT, ToT, Const-o-T, MCTS, and MCTS with CoT. These baselines represent different levels of reasoning structure and search integration, from unguided code generation to exploration guided by unconstrained reasoning. They are included to isolate the contributions of explicit constraint grounding and search-space pruning in CAD design.

**Metrics.** Following the evaluation protocol from Alrashedy et al. (2025), we assess: (i) the geometric alignment between generated and ground-truth objects using the Hausdorff distance (after optimal rotation alignment using the Iterative Closest Point (ICP) algorithm) (Besl & McKay, 1992); and (ii) the success rate, defined as the percentage of code generations that compile and render a valid 3D object. For failed generations, we penalize with a maximum Hausdorff distance of $\sqrt{3}$, which corresponds to the longest possible diagonal in a unit cube.

**Results.** The results in Table 2 demonstrate that MCTS with Const-o-T achieves the best performance across both LLaMA and GPT-4, with the lowest Hausdorff distances (0.343 and 0.302) and the highest success rate of 95.5% for GPT-4. It consistently outperforms standalone CoT, ToT, and MCTS with CoT, highlighting the advantage of combining constraint-based reasoning with tree-based search for CAD code generation.

Table 2: For CAD code generation, we report median Hausdorff distance with IQR and success rate. For math arithmetic, we report accuracy.

| Method | CAD Code Generation (CADPrompt) | | | | Math Arithmetic (GSM8K) | |
| | LLaMA-3.3-70B | | GPT-4 | | LLaMA-3.3-70B | GPT-4 |
| | Hausdorff Dist. | Success Rate | Hausdorff Dist. | Success Rate | Accuracy | Accuracy |
|---|---|---|---|---|---|---|
| CoT | **0.369 (0.553)** | 88.5% | **0.322 (0.442)** | 92.0% | 91.9% | 95.1% |
| CoT + RS | 0.388 (0.504) | **91.5%** | 0.327 (0.471) | 92.5% | 92.4% | 93.1% |
| Const-o-T | 0.385 (0.518) | 89.5% | 0.332 (0.457) | **94.5%** | **92.5%** | **96.1%** |
| ToT | 0.359 (0.500) | **94.0%** | 0.314 (0.439) | 92.0% | 92.6% | 93.7% |
| MCTS | 0.388 (0.534) | 89.5% | 0.332 (0.457) | 94.5% | 92.9% | 95.0% |
| MCTS with CoT | 0.357 (0.498) | 90.5% | 0.316 (0.423) | 93.0% | 91.8% | 95.1% |
| MCTS with Const-o-T | **0.343 (0.465)** | 92.0% | **0.302 (0.479)** | **95.5%** | **93.4%** | **96.2%** |

## 4.3 MATH ARITHMETIC

**Baseline.** We compare CoT, Const-o-T, ToT, MCTS, and MCTS with CoT against MCTS with Const-o-T. This set covers approaches ranging from raw answer prediction to multi-step reasoning and search-based exploration without constraint checks. They are included to highlight the importance of combining structured constraints with search for accurate multi-step arithmetic.

**Metrics.** We evaluate this task using accuracy alone, since each problem has a single correct answer that is always an integer. A prediction is marked correct only if the model's final output exactly matches the ground truth integer.

**Results.** The results on Table 2 shows that integrating structured reasoning significantly improves accuracy across models. The best performance is achieved by MCTS with Const-o-T, reaching 93.4% accuracy with LLaMA-3.3-70B and 96.2% with GPT-4. This outperforms simpler baselines like standalone CoT and CoT+RS, suggesting that combining constraint-guided reasoning with search-based selection is highly effective for arithmetic reasoning. Notably, Const-o-T also performs competitively on its own.

## 5 ANALYSIS

**Search-Space Reduction.** A key advantage of Const-o-T lies in its ability to reduce the effective branching factor of tree search. We empirically measured the average branching factor across search steps for GPT and LLaMA for the Risk domain. Figure 4 demonstrates the results. For GPT-4, the branching factor begins above 140 and gradually decreases for all the approaches, however, MCTS with Const-o-T notably decreases the branching factor by more than 20 compared to MCTS and MCTS with CoT in the first six steps, which rapidly prunes the search space. Similarly, for LLaMA, MCTS with Const-o-T lowers the branching factor by more than 25 between steps three and five.

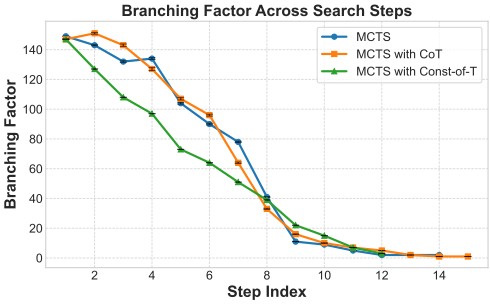 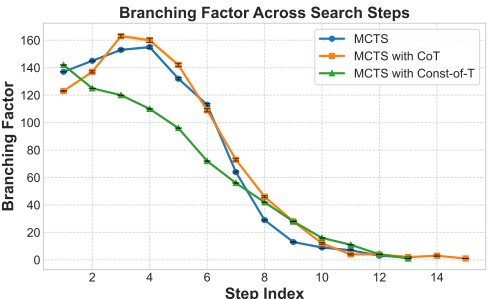

Figure 4: Branching factor with error bars across search steps for GPT-4 (left) and LLaMA-3 (right).

**Efficiency Gains from Constraint-Guided Search.** Figure 5 presents the average wall-clock time per example for GPT-4 and LLaMA 3.3 across three search variants: MCTS, MCTS with CoT, and MCTS with Const-o-T. Although CoT enhances reasoning quality, it comes at a substantial computational cost—for instance, GPT-4's inference time increases dramatically from 20.92s (MCTS) to 53.68s (MCTS with CoT). Const-o-T, however, delivers competitive performance while markedly reducing runtime, decreasing GPT-4's latency to 29.90s and LLaMA 3.3's to merely 9.41s. These findings demonstrate Const-o-T's effectiveness in performing more efficient exploration, resulting in faster and more scalable planning while preserving solution quality.

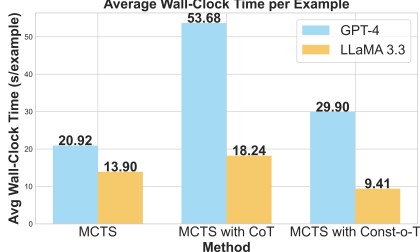

Figure 5: Average inference time per example for GPT-4 and LLaMA 3.3 across three approaches.

**Statistical Analysis.** We conducted one-way ANOVA tests evaluating how performance varies across different experimental settings for GPT-4. For search methods (MCTS, CoT, and Const-o-T), we observe a significant effect on both accuracy ($F(2, 3139) = 4.23$, $p = 0.0147$, $d = 0.052$) and F1-score ($F(2, 3139) = 34.7$, $p < 0.001$, $d = 0.148$), indicating that the choice of search strategy substantially influences model performance, particularly with respect to F1-score. In contrast, prompting strategies show a significant effect on accuracy ($F(2, 3156) = 11.9$, $p < 0.001$, $d = 0.087$) but not on F1-score ($F(2, 3156) = 1.09$, $p = 0.335$, $d = 0.026$), suggesting that while different prompting approaches can shift accuracy, they do not lead to meaningful differences in precision–recall balance. Overall, these results highlight that constraint-guided search plays a stronger role in shaping model quality than prompting variation.

## 6 DISCUSSION

**From Rationales to Controllers: Distinguishing Const-o-T from CoT.** While CoT provides unconstrained reasoning traces that serve primarily as explanatory rationales, Const-o-T reinterprets these intermediate steps as actionable controllers that directly constrain the search process. CoT expands reasoning linearly, often leading to exploration of redundant or infeasible paths, whereas Const-o-T prescribes a feasible solution space by pruning actions inconsistent with symbolic constraints. This shift reframes reasoning from being descriptive to prescriptive: instead of narrating a

possible trajectory, the model commits to verifiable constraints that guide planning toward semantically valid outcomes. This distinction is summarized in Table 7, which directly compares Const-o-T against CoT across representation, purpose, efficiency, and robustness.

**Cross-Domain Generalization.** Our findings show that constraint-guided reasoning consistently improves task performance across diverse domains. In the Risk game, constraints shrink the combinatorial action space by discarding infeasible paths; in CAD code generation, geometric rules ensure that each step respects structural feasibility; and in arithmetic reasoning, constraints enforce equation-level correctness. Together, these results highlight that Const-o-T can act as a domain-agnostic structural prior, providing both the theoretical foundation (See § 5) and empirical improvements (See § 4) observed in our study. Beyond these evaluated settings, the approach holds promise for safety-critical applications such as medical treatment planning and autonomous systems, where reducing hallucinations and guaranteeing valid outcomes are paramount.

**Limitations.** Const-o-T's effectiveness depends heavily on the quality of initial constraint extraction, which remains a key challenge for future work. When LLMs misinterpret user input or generate incomplete symbolic constraints, subsequent search can be misdirected toward suboptimal solutions. Additionally, the framework assumes users can articulate intent in natural language; however, for complex scenarios, users may struggle to express nuanced preferences, potentially leading to constraint extraction failures and human-AI alignment issues.

## 7 RELATED WORK

**Planning and Decision Making.** Classical AI planning relies on heuristic search (Pearl, 1984), probabilistic models such as MDPs (Puterman, 2014), and simulation-based methods like MCTS (Browne et al., 2012), but these approaches require explicit domain modeling (Chakraborti et al., 2020). Recent work explores LLMs as flexible planners, leveraging pretrained knowledge to generate reasoning traces and actions in open-ended settings. Examples include enhancing planning with tree search (Light et al., 2025), combining heuristic reasoning with symbolic search (Saha et al., 2024), and using episodic memory for long-horizon strategies (Zhu et al., 2024). These efforts motivate hybrid frameworks integrating LLMs with classical search, as we do with MCTS.

**Reasoning with LLMs.** LLMs improve performance by reasoning before answering. CoT (Wei et al., 2022) and ToT (Yao et al., 2023) generate step-by-step traces that aid tasks such as arithmetic (Cobbe et al., 2021), commonsense (Zhou et al., 2020), and decision-making (Huang et al., 2022). However, these traces mainly provide rationales without enforcing constraints, often drifting from intent in combinatorial spaces. Our Const-o-T framework addresses this gap by guiding search, enforcing validity, and reducing hallucinations in complex planning tasks.

**Constraint-Guided Reasoning.** Constraints have long ensured feasibility in symbolic planning (Russell et al., 1995), and recent work extends this to LLMs through structured prompting in program synthesis (Austin et al., 2021), CAD generation (Alrashedy et al., 2025), and intent-to-constraint translation (Tambwekar et al., 2023). Other studies show constraints reduce hallucinations in reinforcement learning and neural planning (Garcia & Fernández, 2012). LLMFP (Hao et al., 2025) further formalizes prompts into verifiable representations. Collectively, these works highlight the importance of constraints for building grounded, trustworthy LLM-based planners.

## 8 CONCLUSION

We introduced Constraints-of-Thought (Const-o-T), a framework that transforms unconstrained natural language reasoning into structured, verifiable constraints for guiding language model planning. By representing each reasoning step as an ⟨intent, constraint⟩ pair, our approach provides actionable control over the search process rather than merely explanatory rationales. Our integration with MCTS demonstrates how constraints can effectively prune infeasible branches, reduce branching factors, and direct exploration toward semantically meaningful actions. Comprehensive evaluation across strategic game (Risk), CAD code generation, and arithmetic reasoning shows consistent improvements in accuracy, structural alignment, and hallucination reduction. These findings suggest that structured constraint extraction represents a promising direction for improving LLM reliability in complex planning tasks.

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

# A  APPENDIX

## A.1 RISK GAME ENVIRONMENT

Risk is a strategy-based game originally developed as a board game. Our online adaptation faithfully preserves its core rules, challenging players in diplomacy, territorial conquest, and conflict resolution. The objective is to achieve world domination across a custom-designed map composed of 5 continents and 21 regions. Players take turns deploying troops and attempting to capture territories from their opponents, with combat outcomes determined by dice rolls.

In our setup, the participant plays as the White player, while two heuristic agents control the Black and Grey factions. The game begins with the participant allocating troops to preferred regions by interacting through an AI-powered chat interface. An AI planner then generates a proposed plan that aligns with the participant's stated intent. The participant can either accept this plan or provide feedback to refine it, as illustrated in Fig. 6.

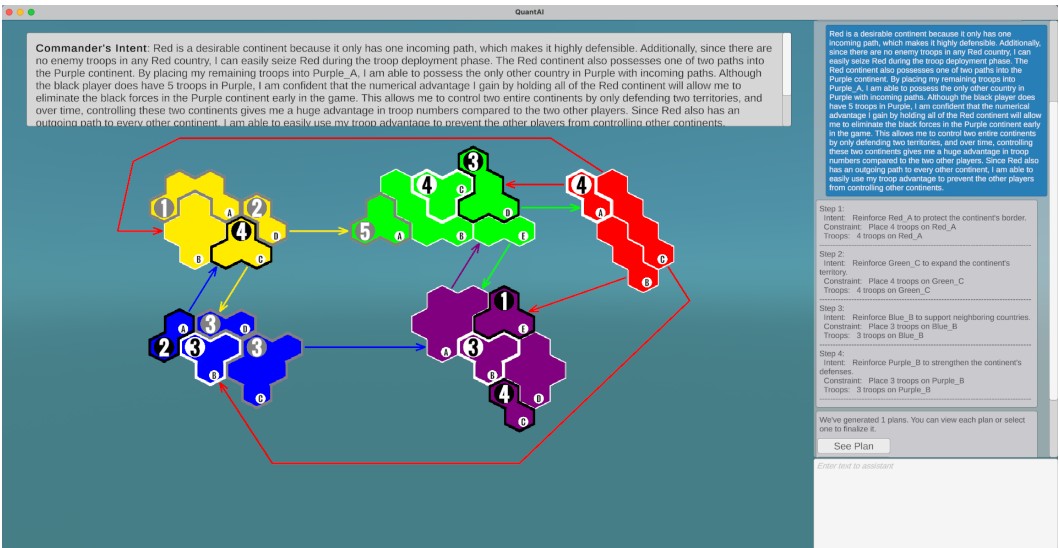

Figure 6: Risk game – Turn 0, showing the initial troop placement by the player.

Once the participant lays down their troops, the opponents make their moves. Once the turn circles back to the participant, as seen in Fig.7, they can either use the same strategy as before or give a new one to the planner.

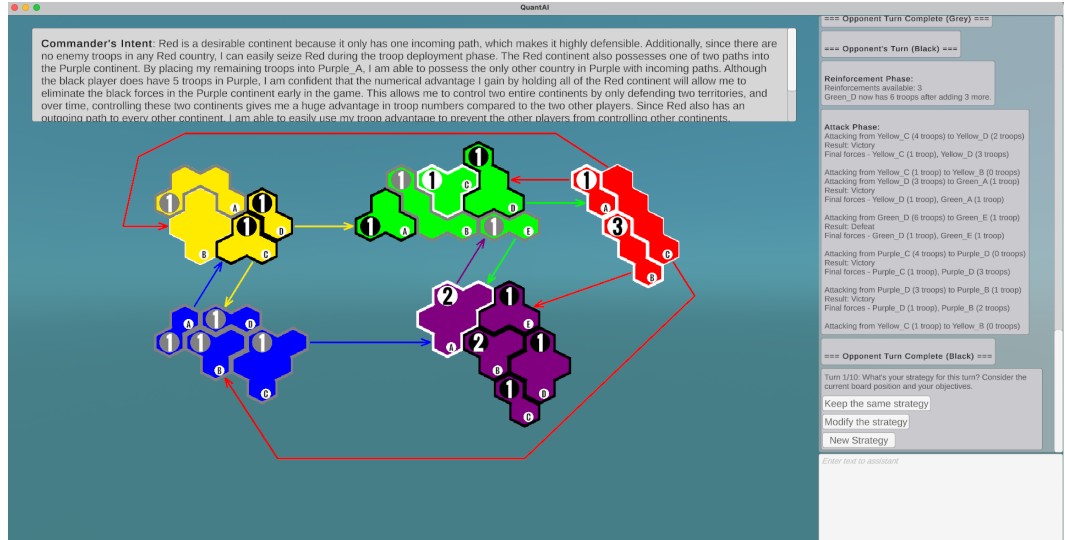

Figure 7: Risk game: Post–Turn 0, as the Black and Gray players make their moves.

This is when the actual turns begin. The participant can choose to Reinforce, Attack, or Freemove (as shown in Fig. 8). At the start of each turn, the player receives reinforcement armies proportional to the number of territories they control, with additional bonus armies granted for holding entire continents. These reinforcements can be used to strengthen key strongholds.

Players may attack adjacent or connected opponent territories—those linked by unidirectional arrows. The outcomes of attacks are determined by dice rolls, with each roll resulting in the loss of a certain number of troops by either the attacker or defender. The more troops committed to an attack, the higher the chances of success. A battle continues until the attacker chooses to stop, runs out of armies to attack with, or successfully eliminates the last defending unit—at which point they take over the territory by moving armies into it.

At the end of the turn, the player may perform a Freemove, redistributing armies between their own connected territories. This cycle repeats until one player achieves world domination.

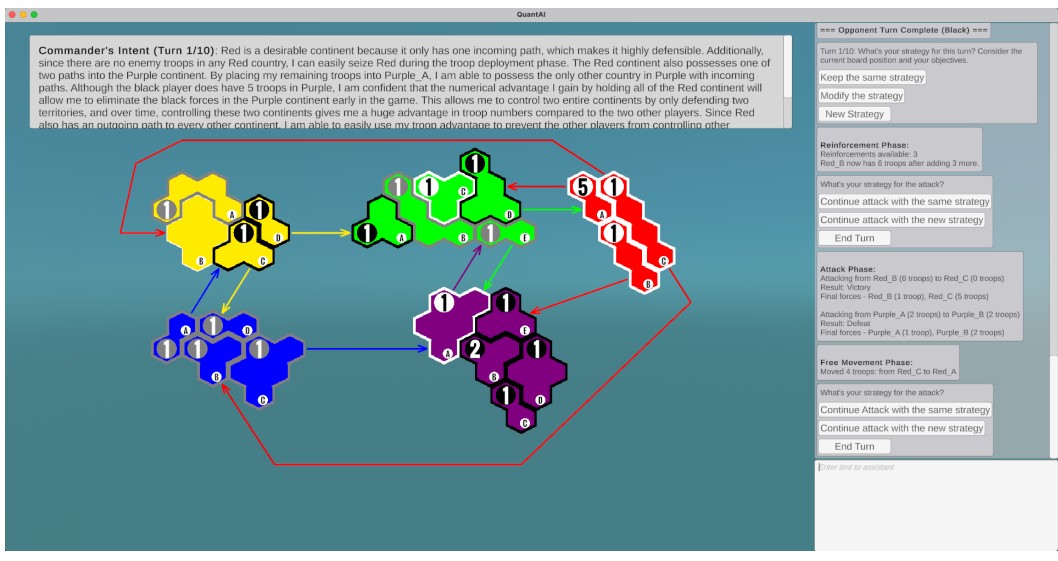

Figure 8: Risk game: Turn 1, where the player proceeds through the reinforcement, attack, and freemove phases.

## A.2 CONSTRAINT-GUIDED MCTS

The algorithm takes as input the user description $D$, the initial state $s_0$, the map $M$, and a constraint sequence $c_t = \langle \text{intent}_t, \text{constraint}_t \rangle$ extracted from Alg. 2. The output is the most optimal path from the root node. In line 3, the total number of MCTS rollouts is set equal to the number of extracted constraints. Each constraint is used to guide one rollout: during this rollout, the language model generates the next candidate action, and MCTS leverages the corresponding constraint for both selection and evaluation (see line 5). During the selection phase (lines 8–10), the most promising node is selected using the Upper Confidence Bound (UCB) formula. In the expansion phase (lines 12–15), the language model proposes several potential next actions, and MCTS filters them by checking legality and constraint satisfaction. The evaluation phase (line 17) employs a domain-specific fitness function (See the detail in Section A.6) to assess the quality of each node. In the Risk game, this function quantifies alignment with the user's strategic intent, satisfaction of symbolic constraints, and progress toward territorial objectives. For CAD code generation and mathematical reasoning, we leverage LLMs to evaluate whether the generated outputs are consistent with the intended design or correct solution. Finally, in the backpropagation step, the evaluation scores are propagated along the selected path back to the root to inform future rollouts.

---

**Algorithm 1** Constraint-Guided MCTS (CG_MCTS)

---

**Require:** User description $D$; Initial state $s_0$; map $M$; constraint sequence $\mathcal{C} = \left[ (\text{intent}_i, \text{constraint}_i) \right]_{i=1}^{K}$ from Alg. 2;
**Ensure:** Plan $\pi$ or root action $a^\star$

1: **function** CG_MCTS($s_0, M, \mathcal{C}$)
2:      Create root node $v_0$ with state $s_0$;
3:      $R \leftarrow |\mathcal{C}| \equiv K$                         ▷ number of rollouts equals number of constraints
4:      **for** $k = 1$ **to** $R$ **do**
5:          $c \leftarrow \mathcal{C}[k]$                      ▷ use the $k$-th constraint to guide this rollout
6:          $v \leftarrow v_0$; path $\leftarrow [\,]$
7:          **while not** TERMINAL($v$) **do**                   ▷ Selection
8:              $a \leftarrow \arg \max\limits_{a \in \mathcal{A}(v)} \left[ Q(v,a) + c_{\text{uct}} + \lambda \cdot \sqrt{\frac{\ln(1+N(v))}{1+N(v,a)}} \right]$
9:              path $\leftarrow$ path $\cup \{(v,a)\}$
10:         $v \leftarrow$ CHILD($v, a$)
11:          **end while**
12:          $\tilde{\mathcal{A}} \sim$ TOPK($P_{LM}(\cdot \mid \text{state}(v), c, D)$, $K_{\text{gen}}$)          ▷ Expansion
13:          $\mathcal{A}_{\text{legal}} \leftarrow \{a \in \tilde{\mathcal{A}} \cap \mathcal{A}(v) : \mathbb{I}_c(a \mid v) = 1\}$
14:          **if** $\mathcal{A}_{\text{legal}} =$ **then** $\mathcal{A}_{\text{legal}} \leftarrow \{a \in \mathcal{A}(v) : \mathbb{I}_c(a \mid v) = 1\}$ **else keep top** $K_{\text{expand}}$
15:          For each $a \in \mathcal{A}_{\text{legal}}$:
16:              $v' \leftarrow$ CREATECHILD($v, a$, NextState($v, a$))
17:          $v \leftarrow$ EVALUATION($v$)                 ▷ Evaluation
18:          BACKUP(path, $V(v)$)               ▷ Backpropagation
19:      **end for**
20:      **return** $a^\star$
21: **end function**

---

## A.3 CONSTRAINTS-OF-THOUGHT (CONST-O-T) ALGORITHMS

The key idea behind Const-o-T is to extract both the intent and the associated constraints from the user's input, and then leverage them to guide large language models (LLMs) or tree search algorithms such as MCTS. These extracted constraints serve two main purposes: (i) they steer the generation process to align with the user's high-level intent, and (ii) they provide a means to verify whether the generated output satisfies domain-specific requirements—such as game legality rules or geometric constraints in CAD generation. This feedback mechanism also enables iterative refinement of the output.

Algorithm 2 illustrates this process: in line 2, the LLM receives the input and generates the corresponding intent-constraint pairs. From lines 4 to 10, the algorithm verifies each extracted constraint, ensuring that downstream reasoning or planning steps adhere to the specified rules.

---

**Algorithm 2** Constraints-of-Thought (Const-o-T) Extraction from user input

---

**Require:** User description $D$; map state $M$; prompt template $\mathcal{T}$
**Ensure:** Sequence $\mathcal{C} = \left[(\text{intent}_i, \text{constraint}_i)\right]_{i=1}^{K}$; total count $K$

1: **function** CONSTCOT_EXTRACT($D, M, \mathcal{T}$)
2:     $\widehat{\mathcal{C}} \leftarrow LM(D, M, \mathcal{T})$
3:     $\mathcal{C} \leftarrow [\,]$
4:     **for all** $c \in \widehat{\mathcal{C}}$ **do**
5:         **if** VALIDATE($c$) **then**
6:             $\mathcal{C} \leftarrow \mathcal{C} \cup [c]$
7:         **end if**
8:     **end for**
9:     $K \leftarrow |\mathcal{C}|$
10:     **return** $(\mathcal{C}, K)$
11: **end function**

12: **function** VALIDATE($c$)
13:     **require** fields present: step_id, intent (natural language), constraint (formal/actionable)
14:     **require** intent length within bounds; constraint matches grammar $\mathcal{G}$ or schema $\mathcal{S}$
15:     **require** constraint feasibility under $X'$ if available (e.g., legal action, resource limits)
16:     **return** true if all checks pass; else false
17: **end function**

---

## A.4 USER STUDY

The user study was conducted with 18 participants in total with 5 female and 13 male participants. All participants over the age of 18 were welcome but the average age of the participants was 21.7. Participants had varying levels of experience with AI systems, gaming, strategy-based games and Risk. Trust scale items were adapted from validated measures of trust in automation (Jian et al., 2000; Kizilcec, 2016) and technology acceptance (Davis, 1989), with wording contextualized to the Risk domain (e.g., "I trust the system's troop placement/plan suggestions"). Usability scale items were also contextualized to the Risk domain (e.g., "I could easily predict how changes in my strategy description would affect the system's decisions.") taking reference from (Liao et al., 2020). The reliability of each metric was calculated using Chronbach's Alpha and reported in Table 3. Normality and heteroskedasticity assumptions were checked using a Shapiro-Wilk test and a Levene's test respectively. Transparency and usability metrics passed these tests. The results of the one way ANOVA test and post-hoc Tukey's HSD test for transparency and usability and the Kruskal-Wallis test and post-hoc Dunn's test for trust and alignment are reported in Table 4 and Table 5 respectively.

## A.5 ABLATION STUDY.

To better understand the contributions of constraint settings and search strategies, we conducted an ablation study on a randomly selected subset of 150 examples using GPT-4. First, we compared soft versus hard constraint enforcement. In the soft setting, the model receives the constraint as optional guidance and may ignore it during generation. In contrast, hard constraints are strictly enforced by rejecting and regenerating outputs until they satisfy the constraint. This shift from lenient to enforced constraint handling improved accuracy from 76% to 87% and F1-score from 0.70 to 0.78. Second, we evaluated the effect of constraint-guided search depth. Allowing unconstrained expansion yielded 82% accuracy and 0.72 F1-score, whereas limiting tree depth to the number of constraints improved performance to 87% accuracy and 0.78 F1-score (See Table 6).

Table 3: Reliability analysis by metric and mode. Alpha > 0.9 has excellent reliability, Alpha > 0.8 has good reliability and Alpha > 0.7 has acceptable reliability

| Metric | Mode | Alpha | Reliability |
|--------|------|-------|-------------|
| | Mode 1 | 0.95 | Excellent |
| Transparency | Mode 2 | 0.93 | Excellent |
| | Mode 3 | 0.95 | Excellent |
| | Mode 1 | 0.81 | Good |
| Usability | Mode 2 | 0.76 | Acceptable |
| | Mode 3 | 0.78 | Acceptable |
| | Mode 1 | 0.79 | Acceptable |
| Trust | Mode 2 | 0.93 | Excellent |
| | Mode 3 | 0.90 | Excellent |

Table 4: ANOVA and post-hoc results for transparency and usability

| | Transparency | Usability |
|--------|--------------|-----------|
| APA | $F(2, 51) = 7.78, \; p < 0.01$ | $F(2, 51) = 10.23, \; p < 0.001$ |
| Shapiro $p$ | 0.427 | 0.971 |
| Levene $p$ | 0.996 | 0.843 |
| Eta$^2$ | 0.234 | 0.286 |
| Power | 0.951 | 0.987 |

**Tukey HSD pairwise comparisons**

| Comparison | Transparency (mean diff, $p$) | Usability (mean diff, $p$) |
|------------|-------------------------------|----------------------------|
| Mode 1 vs Mode 2 | -1.21, $p = 0.001$ ** | -1.83, $p = 0.0001$ *** |
| Mode 1 vs Mode 3 | -0.84, $p = 0.026$ * | -1.07, $p = 0.030$ * |
| Mode 2 vs Mode 3 | +0.36, $p = 0.481$ n.s. | +0.76, $p = 0.156$ n.s. |

**Cohen's $d$ (pairwise effect sizes; d > 0.8 signifies large effect size)**

| Comparison | Transparency | Usability |
|------------|--------------|-----------|
| Mode 1 vs Mode 2 | 1.29 | 1.52 |
| Mode 1 vs Mode 3 | 0.90 | 0.89 |
| Mode 2 vs Mode 3 | -0.39 | -0.61 |

## A.6 FITNESS FUNCTION

For the game of *Risk*, during the troop deployment phase, we evaluate a candidate deployment state $v$ using a fitness function that balances goal satisfaction against constraint violations. The overall fitness is defined as

$$V(v) \; = \; \sum_{i=1}^{6} w_i \, g_i(v) \; - \; \lambda \sum_{m=1}^{9} c_m(v), \tag{5}$$

where $g_i(v)$ are normalized goal scores, $w_i$ are non-negative weights that encode the relative importance of each goal, and $c_m(v)$ are binary indicators of constraint violations. The penalty coefficient $\lambda$ is set sufficiently large to ensure that any violation dominates the weighted sum of goals, thereby prioritizing feasible deployments over infeasible ones.

Table 5: Kruskal-Wallis and post-hoc results for transparency and usability

|  | Trust | Alignment |
|---|---|---|
| APA | $H(2) = 19.80, \ p < 0.001$ | $H(2) = 23.75, \ p < 0.001$ |

**Dunn pairwise comparisons (adjusted $p$)**

| Comparison | Trust | Alignment |
|---|---|---|
| Mode 1 vs Mode 2 | $p = 0.0001$ *** | $p < 0.00001$ *** |
| Mode 1 vs Mode 3 | $p = 0.0014$ ** | $p = 0.0011$ ** |
| Mode 2 vs Mode 3 | $p = 1.000$ n.s. | $p = 0.814$ n.s. |

**Cohen's $d$ (pairwise effect sizes; d > 0.8 signifies large effect size)**

| Comparison | Trust | Alignment |
|---|---|---|
| Mode 1 vs Mode 2 | 1.76 | 2.19 |
| Mode 1 vs Mode 3 | 1.51 | 1.67 |
| Mode 2 vs Mode 3 | -0.26 | -0.43 |

Table 6: Ablation study results.

| Method | Experiment | Acc. | F1 |
|---|---|---|---|
| Constraint | Soft | 76% | 0.70 |
|  | Hard | **87%** | **0.78** |
| Search | Unconstrained | 82% | 0.72 |
|  | Guided (Depth) | **87%** | **0.78** |

**Goals.** The goal functions $g_i(v)$ measure how well the player's configuration $v$ satisfies strategic objectives:

$$g_1(v) = \frac{\#\{\text{occupied territories adjacent to enemy}\}}{\#\{\text{occupiable territories adjacent to enemy}\}}, \tag{6}$$

$$g_2(v) = \frac{\#\{\text{countries controlled}\}}{\#\{\text{countries occupiable}\}}, \tag{7}$$

$$g_3(v) = 1 - \frac{1}{|\mathcal{T}(v)|(|\mathcal{T}(v)| - 1)} \sum_{\substack{u,w \in \mathcal{T}(v) \\ u \neq w}} \frac{d(u,w)}{d_{\max}}, \tag{8}$$

$$g_4(v) = \frac{\#\{\text{troops adjacent to enemy}\}}{\#\{\text{total troops}\}}, \tag{9}$$

$$g_5(v) = \frac{\#\{\text{border troops on controlled continents}\}}{\#\{\text{troops on controlled continents}\}}, \tag{10}$$

$$g_6(v) = 1 - \frac{\#\{\text{unique enemy players adjacent}\}}{\#\{\text{maximum enemies}\}}. \tag{11}$$

*Descriptions.* $g_1$: surround enemy territories; $g_2$: maximize territorial control; $g_3$: minimize average pairwise troop distance (uses graph distance $d(\cdot, \cdot)$, normalized by $d_{\max}$); $g_4$: maximize battles throughout the game; $g_5$: fortify borders of continents you control; $g_6$: limit exposure to many enemies.

**Constraints.** Constraints are formulated as binary functions that equal 1 when violated and 0 otherwise:

$$c_1(v) = \mathbb{1}[\text{no troop on required continent}], \qquad c_2(v) = \mathbb{1}[\text{troop placed on forbidden continent}], \tag{12}$$

$$c_3(v) = \mathbb{1}[\text{cannot reach continent in one move}], \qquad c_4(v) = \mathbb{1}[\text{border of continent not defended}], \tag{13}$$

$$c_5(v) = \mathbb{1}[\text{insufficient troops to defend continent}], \qquad c_6(v) = \mathbb{1}[\text{fewer than required countries}], \tag{14}$$

$$c_7(v) = \mathbb{1}[\text{troops on fewer than required continents}], \quad c_8(v) = \mathbb{1}[\text{fewer than required troops per country}], \tag{15}$$

$$c_9(v) = \mathbb{1}[\text{troops on more than allowed continents}]. \tag{16}$$

Together, these definitions create a fitness landscape in which higher scores correspond to strategically advantageous and constraint-compliant configurations. The weighting scheme allows tailoring the optimization toward different strategic preferences while ensuring that hard constraints remain non-negotiable.

### A.7 COMPARISON OF CHAIN-OF-THOUGHT (COT) AND CONSTRAINTS-OF-THOUGHT (CONST-O-T)

CoT relies on unconstrained natural language reasoning that helps describe the thought process but lacks verifiability and control over the search process. It's best suited for single-step QA or explanation tasks, but often leads to large search spaces and hallucinations.

In contrast, Const-o-T converts strategies into structured symbolic constraints, which can be formally verified and used to guide or prune search (e.g., in MCTS). This results in more efficient, robust, and goal-aligned planning, making it especially effective for multi-step tasks like strategy games and CAD design.

Table 7: Comparison of Between Chain-of-Thought (CoT) and Constraints-of-Thought (Const-o-T).

| Aspect | Chain-of-Thought (CoT) | Constraints-of-Thought (Const-o-T) |
|---|---|---|
| Representation | unconstrained natural reasoning trace | Structured symbolic constraints (equations, rules) |
| Purpose | Describes reasoning steps | Prescribes feasible solution space |
| Search Interaction | Linear expansion; may increase branching | Guides/prunes expansions; controls horizon and branching factor |
| Verification | No formal mechanism; correctness by final output | Constraints checkable via solvers/game rules |
| Efficiency | Larger search space; redundant paths possible | Compressed space; faster convergence |
| Robustness | Prone to hallucinations and drift | Enforces consistency; fewer invalid solutions |
| Use Case Fit | Best for single-step QA/explanation | Suited for multi-step planning, strategy, CAD/game tasks |
| Novelty | Thoughts as *rationales* | Thoughts as *controllers* (constraints for symbolic search) |

### A.8 EXAMPLED OF COMMANDER'S INTENT DATASET.

This example from the Commander's Intent dataset highlights the complex strategic reasoning required in a Risk board game scenario (See Figure 9). The natural language strategy demonstrates multi-step task planning, where the player first analyzes the current board state and then formulates a plan. The accompanying map depicts the actual game state, with the ground truth showing optimal

troop placements of seven units each in Red_B and Red_C. These placements align with the strategic focus on securing the Red continent, as outlined in the textual strategy.

---

**Language strategy.**

This one was difficult but I determined that controlling red would allow me the greatest chance of success as it provides a good base for defense while also granting large movement opportunities to attack most of the board. Green will be the biggest challenge as they have the most number of troops with 20 that they can move around, so the strategy would be to gain control of red as quickly as possible before moving in on yellow while leaving the borders of red as strong as possible. Gaining control of yellow and red would allow me to keep green on the defense from opposite sides, but give me access to work control of blue as green will most likely make moves to gain purple as a stronghold. The trick will be playing a long game to maintain control of the split continents while green will ultimately control the center of the board until I can attempt to force them down into the purple region and attack from several sides.

---

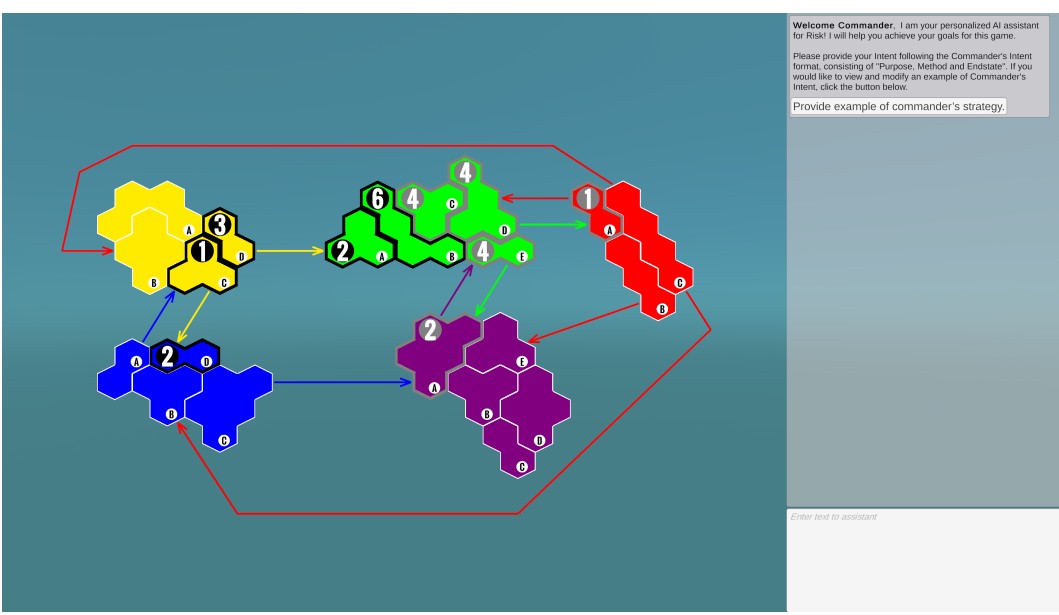

Figure 9: An example from the Commander's Intent dataset. Top: a natural language strategy description. Bottom: the corresponding map scenario. The ground truth troop placements are Country = Red_B with 7 troops and Country = Red_C with 7 troops.

## A.9 CONSTRAINT-OPTIMIZATION BASELINE FOR TROOP PLACEMENT

We also implemented a constraint optimization based baseline. It is a two-step strategy. In the first step, the commander's intent is provided to a large language model (we use ChatGPT 4.0 mini) to identify the set of active constraints. These constraints are then enforced using Google's OR-Tools library (Google LLC, 2025) to trim the search space and obtain a feasible set of troop deployments. If the constraints predicted by the LLM yield no feasible solutions, we retain the largest subset of constraints that produces a feasible solution.

In the second step, we evaluate all deployments in the feasible set using only the goal component of the fitness function:

$$V(v) = \sum_{i=1}^{6} w_i \, g_i(v), \tag{17}$$

where the weights $w_i$ are provided by the LLM, derived from the commander's intent expressed in natural language, to reflect strategic priorities. The deployment with the highest value of $V(v)$ is then selected as the baseline solution.

The performance of the constraint-optimization baseline is shown in Table 8. Even though it leverages constraint-based optimization, its performance compared to human provided troop placement ground-truth is significantly worse than our method (see Section 4). In addition, it is not

| Method | F1 (%) | Accuracy (%) |
|---|---|---|
| Constraint-Optimization Baseline | 57.0 | 62.8 |

Table 8: Results of the constraint-optimization baseline.

very flexible: each new domain requires hand-crafting a constraint model and solver formulation, whereas our approach transfers more easily and remains flexible across a wide range of applications.

### A.10 EXAMPLES OF CAD CODE GENERATION FROM CADCODEPROMPT DATASET.

This subsection presents representative example from CADCodePrompt dataset to illustrate the natural language descriptions paired with their corresponding CAD code implementations (See Figure 10). The example demonstrates how natural language of CAD design can be translated into executable Python code using CADQuery.

Write Python code using CADQuery to create a triangular 3D object. First, draw a sketch of an equilateral triangle, pointing downwards. Next, cutout a semicircle from the bottom corner of the triangle. The diameter of this semicircular cutout should be approximately 2/3rd of the length of each side of the triangle. Finally, extrude this sketch to create a 3D object.

(a) The natural language descriptions of the 3D object.

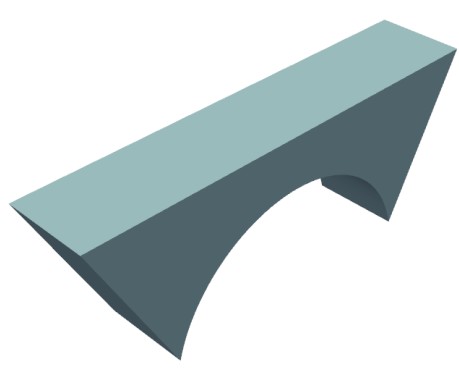

(b) CAD object

```python
import cadquery as cq
from typing import List,Tuple

length:float = 0.929516
width:float = 0.54
height:float = 0.3
top_length:float = 1.5

top_x:float = (length - top_length)/2

points:List[Tuple[float, float]] = [
    (0, 0),
    (length, 0),
    (top_x + top_length, width),
    (top_x, width),
]

work:cq.Workplane = (
    cq.Workplane("XZ")
    .center(-length/2,-width/2)
    .polyline(points).close()
)

part:cq.Workplane = work.extrude(height).translate((0,height/2,width
    /2+0.12))

radius = (length/2)+0.014

ellipse = cq.Workplane("XZ").circle(radius).extrude(height).translate
    ((0,height/2,))
part = part.cut(ellipse.translate((0,0,width+0.12-radius-0.18)))

cq.exporters.export(part, 'Code_Ground_Truth.stl')
```

(c) Python Code

Figure 10: An example from the *CADPrompt* dataset, showing (a) the prompt, (b) the corresponding CAD object, and (c) the human-annotated Python code used to generate the CAD object.

---

**Constraints-of-Thought Prompt for Strategic Planning in the Risk Game**

You are a **strategic assistant for the Risk board game**. Your task is to **immediately allocate all troops** according to the Commander's intent, using step-by-step reasoning based on Constraints-of-Thought (Const-o-T) and Monte Carlo Tree Search (MCTS) decision-making.

**Input Provided:**

- A natural language description of the Commander's intent for this turn's troop placement (e.g., "Fortify the Green continent and strengthen borders.").
- The current map status (countries, ownership, and unoccupied territories).

**Your Output:**

- Generate a sequential and complete plan for troop placement **for this turn only**.

For **each step**, provide:

- A concise, natural-language **intent** (why the troop is being placed in that country)
- A **formal placement constraint** (e.g., "Place 3 troops on Green_C")
- The exact country and number of troops as: `["Green_C", 3]`
- The Placement must include a valid country and a numeric troop count; the troop count cannot be None, empty, or non-numeric. Only output a numeric value for the number of troops.

**Example Constraints:**

- Place 'n' troops on Country 'X'
- Attack Country 'X' from Country 'Y' with 'n' troops
- Move 'n' troops to Country 'X' from Country 'Y'
- Add 'n' troops to reinforce Country 'X'

**Game Environment:**
Risk is a board game in which an army commander tries to take over the world by defeating all enemy troops and controlling all countries. Risk is a simplified version of real conflict, and has rules to reflect this:

- Players control countries by having troops in them.
- The more countries and continents a player controls, the more resources they get.
- Players win countries from other players by battling with their troops.
- The more troops a player has when battling, the more likely they are to win.
- Players can only attack or be attacked by countries that are next to them.
- Some map connections are one-way only.

Our modified RISK Map contains 5 continents - Red, Green, Purple, Yellow and Blue. Each continent is made up of countries. Red continent has 3 countries, Green has 5 countries, Purple has 5 countries, Yellow has 4 countries and Blue has 4 countries. Green_A, Yellow_B, Blue_C, etc. are referred to as countries or territories Green, Yellow, Blue, Red, Purple are referred to as continents. Continents also have different connections between them through which the troops can move. These connections are one way i.e troops from the source country can only move to the destination country and not the other way round. The map has the following connections - Yellow_D is connected to Green_A, Greed_D is connected to Red_A, Red_A is connected to Green_D, Red_B is connected to Purple_E, Red_C is connected to Yellow_B, Red_C is connected to Blue_B, Blue_A is connected to Yellow_C, Yellow_C is connected to Blue_D, Blue_C is connected to Purple_A, Purple_A is connected to Green_E and Green_E is connected to Purple_A.

**INPUT**
**Commander's Intent:** {Strategy_Description}
**The Map Status:** {mapStatus}
You may place troops only in countries that are unoccupied according to the current map status.

**OUTPUT FORMAT (JSON-like):**

```
Constraint-of-thoughts [
  {
    "step_id": 1,
    "intent": "Reinforce Green_C to protect the continent's border.",
    "constraint": "Place 5 troops on Green_C",
    "placement": ["Green_C", 5]
  }
]
```

**INSTRUCTIONS**

- Focus only on troop placement this turn. Do not suggest attacks, moves, or future planning.
- Use formal placement constraints for each action.
- Do not select the same country more than once.
- Be concise and ensure placements clearly support the Commander's current intent.
- Output the full CoT sequence in the specified format.

**Now, reason step-by-step and output your immediate troop placement sequence.**

Figure 11: Prompt Template: Constraints-of-Thought for Risk Game Troop Placement

---

**Constraints-of-Thought Prompt for CAD code generation**

**You are a symbolic CAD modeling assistant.** Your task is to generate a 3D CAD object by reasoning step-by-step using **Constraints-of-Thought (Const-o-T)** — a structured form of geometric planning based on user instructions.

You will be given:

- A natural language description of a 3D object

**Your output must consist of a sequence of steps**, where each step includes:

1. A short natural-language **intent** (what to do and why)

2. A plain-English geometric **constraint** (e.g., "The base should be a box with width 1.0, depth 0.75, and height 0.25", or "The hole should be centered on the base, with radius 0.1, and aligned along the Z-axis")

Each step must logically build upon previous ones.

**INPUT:**
**Natural Language Description:** {language_description}

**Output format (JSON-like):**

```
{Constraints-of-Thought[
  {"Step_id": 1,
   "Intent": "Create the base plate.",
   "Constraint": "Make a rectangular box with width 1.0 units,
   depth 0.75 units, and height 0.25 units."},
  {"Step_id": 2,
   "Intent": "Add a centered hole.",
   "Constraint": "Drill a circular hole with radius 0.1 units at the center of the base,
   oriented along the Z-axis."}
]}
```

**INSTRUCTIONS:**

- Think like a constraint solver: extract **clear symbolic relationships** but describe them in English.
- Use **plain sentences** that can be easily mapped to CAD operations.
- Be concise. Avoid unnecessary primitives.
- The final object must follow all described constraints.

**Now reason step-by-step and output the full Constraints-of-Thought (Const-o-T) sequence.**

Figure 12: Prompt for symbolic 3D CAD modeling using Constraints-of-Thought (Const-o-T).

---

**Evaluation Prompt for Math Arithmetic Step**

You are an evaluation agent tasked with assessing whether a reasoning step effectively contributes to solving a math problem.

Your job is to evaluate a single step in the problem-solving process and return a score between 0 and 1, where:

- - 1 means the step is highly useful and directly helps solve the problem,
- - 0 means the step is irrelevant, misleading, or incorrect,
- - Intermediate values (e.g., 0.5) indicate partial usefulness or vague contribution.

**INPUT:**

- **Problem:** {question}
- **Step to evaluate:** {Step}

**Output format (JSON):**

```
{"score": float between 0 and 1}
```

**Now assess the step and output the score.**

Figure 13: Prompt for evaluating math arithmetic step in MCTS.

---

**Evaluation Prompt for CAD code Step**

You are an evaluation agent tasked with assessing whether a **generated CAD modeling code snippet** effectively contributes to building the intended 3D model.

Your job is to evaluate a single step of **CAD code** within the Constraints-of-Thought pipeline and return a score between 0 and 1, where:

- - 1 means the code is highly useful, precise, and directly implements the intended modeling operation,
- - 0 means the code is irrelevant, incorrect, or harmful to the design,
- - Intermediate values (e.g., 0.5) indicate partial correctness, vague implementation, or incomplete alignment with the intended step.

**INPUT:**

- **CAD Object Description:** {question}
- **CAD Code to evaluate:** {Step}

**Output format (JSON):**

```
{"score": float between 0 and 1}
```

**Now assess the step and output the score.**

---

Figure 14: Prompt for evaluating CAD code step in MCTS.

---

**Constraints-of-Thought Prompt for math arithmetic**

**You are a symbolic math solver.** Your task is to solve a math word problem by reasoning step-by-step using **Constraints-of-Thought (Const-o-T)** — a structured form of logical planning.

You will be given:

- A natural language word problem involving arithmetic reasoning.

**Your output must consist of a sequence of steps**, where each step includes:

1. A short natural-language **Intent** (what is being computed and why).
2. A precise **Constraint** using symbolic math expressions (e.g., `x = 2 * y, total = cost_per_egg * eggs_sold`).

**INPUT:**
**Word Problem:** {question}
**OUTPUT FORMAT (JSON-like):**

```
json{Constraint-of-thoughts [
  {
    "Step_id": 1,
    "Intent": "Determine the number of bolts of blue fiber needed for one robe.",
    "Constraint": "blue_bolts = 2"
  },
  {
    "Step_id": 2,
    "Intent": "Calculate the number of bolts of white fiber needed,
    which is half as much as blue fiber.",
    "Constraint": "white_bolts = blue_bolts / 2"
  },
  {
    "Step_id": 3,
    "Intent": "Add the number of blue and white bolts to find
    the total number of bolts needed.",
    "Constraint": "total_bolts = blue_bolts + white_bolts"
  }
]}
```

**INSTRUCTIONS:**

- Use precise arithmetic operations for each step (addition, subtraction, multiplication, division).
- Define and name variables explicitly to reflect the quantities being calculated (e.g., `total_bolts`, `num_apples`, `cost_per_item`).
- Each step should logically follow from the previous one, building upon the calculations.
- Only use the information given in the problem — no assumptions or outside knowledge.
- Your solution must contain **no more than 7 steps**. Merge or skip trivial operations when appropriate.

---

Figure 15: Prompt for math arithmetic using Constraints-of-Thought (Const-o-T).

