# OpenReview forum: "Constraints-of-Thought: A Framework for Constrained Reasoning in Language-Model-Guided Search"
_ICLR.cc/2026/Conference — ICLR 2026 Conference Withdrawn Submission_

### Official Review · Reviewer_9dNA · 2025-10-22

**Soundness:** 3
**Presentation:** 3
**Contribution:** 3
**Rating:** 4
**Confidence:** 2

**Summary:**

The paper proposes Constraints-of-Thought (Const-o-T). The method represents each reasoning step as an ⟨intent, constraint⟩ pair and uses these pairs to guide MCTS so search stays on semantically valid, user-aligned paths. Constraints prune branches and also shape selection via a UCB term augmented with an LLM log-probability; rollout budgets scale with the number of extracted constraints. The approach is evaluated on Risk (initial troop placement), CAD code generation (CADPrompt), and GSM8K arithmetic, with a small user study in Risk reporting higher transparency/usability/trust/alignment in the “aligned” mode.

**Strengths:**

- Clear, structured intermediate representation. The ⟨intent, constraint⟩ abstraction is concrete, actionable, and generally novel; constraints define a reduced feasible action set A'(s), enabling verifiable checks and pruning during search.
- Human-factors and statistical significance: The user study (n=18) finds statistically significant improvements in transparency, usability, trust, and alignment for the “aligned” mode. The confidence intervals provided are also very informative as are the statistical tests for trust and alignment and transparency
- Ablations and efficiency. Reported reductions in branching factor and runtime (vs. MCTS / MCTS+CoT), plus an ablation of soft vs. hard constraint enforcement.
- The planning and search domains which are tested on are very interesting: for example Risk, CAD code for 3D model generation
- Search integration beyond post-hoc verification. Constraints influence selection (modified UCB), expansion (legality/constraint filters), and rollout budgeting—not just output filtering—so the framework is prescriptive rather than descriptive.

**Weaknesses:**

- Methodological novelty is incremental. The pipeline largely composes off-the-shelf pieces—LLM-extracted constraints, standard MCTS, and a UCB tweak—applied to three benchmarks. Prior work already translates language to constraints (e.g., optimization-oriented prompts/constraint synthesis) such as [1] and integrates LLMs with search; here the main step is the naming + consistent use of ⟨intent, constraint⟩ within MCTS.
- Constraint quality is under-measured. We see the effects of enforcement mode and depth policy, but not the accuracy/recall of the extracted constraints themselves or how bad constraints degrade performance. Given the centrality of constraints, reporting their precision/recall (and outcome sensitivity to noisy/partial constraints) would make the results more robust. It’s hard to tell whether gains come from better search or simply lucky, high-quality constraints.
- There are more recent agentic strategies such as Self Discover [2] and LATS [3], it would have been interesting to see experiments on those for greater coverage.
- Similar to the prior note strong settings commonly reported in the literature (e.g., self-consistency sampling over CoT/ToT, temperature changes, or search with simple constraint filters but different selection rules) are not discussed. Doing further experimentation or discussing such changes may be helpful for understanding the performance gains of the method and seeing if they still hold under the aforementioned settings
- Theoretical framing is light. While the paper formalizes the POMDP and gives an MCTS-compatible scoring, there’s no search-complexity or regret bound showing when constraints provably help (e.g., bounding expected branching factor reduction as a function of constraint accuracy).


[1] https://arxiv.org/pdf/2410.24175v2

[2] https://proceedings.neurips.cc/paper_files/paper/2024/file/0db7f135f6991e8cec5e516ecc66bfba-Paper-Datasets_and_Benchmarks_Track.pdf

[3] https://arxiv.org/abs/2402.17161

**Questions:**

- Constraint quality: How often are extracted constraints wrong/incomplete and how do you detect/repair them online? (Your ablation hints at “hard vs soft constraints”; more diagnostics would help.)
- Computational complexity & budget scaling. Could you characterize the time/sample complexity of your constrained search, at least empirically and with compute time on your domains? For example, report the effective branching factor and nodes expanded as a function of (i) the number of extracted constraints, (ii) constraint accuracy (precision/recall), and (iii) the rollout/visit budget—on Risk and CAD. See [1] [2]. For [2] can you briefly discuss how your method differs?
- Do you perform “on-the-fly” code repair? If so, what’s the cost model? Some recent search-with-feedback methods refine generated code/thoughts during search using execution or verifier feedback, and they discuss the iteration cost and its effect on total complexity. If your CAD/Risk pipelines ever re-generate or repair snippets (e.g., when constraints fail or code doesn’t compile), please quantify: average repair iterations per solved instance, success rate per iteration, and the impact on total node expansions / wall-clock. See [3] [4]
- Generalization: Do results hold if constraints are partially hidden / perturbed (e.g., renamed fields, schema noise) to test robustness beyond surface patterns?
- Failure modes by domain. For Risk, CAD, and GSM8K, can you categorize the top failure types (e.g., wrong/omitted constraint, legal but low-value action, numeric slip) and quantify their frequencies?
- Generalization of the representation. Beyond these domains, what task properties (it is difficult to extrapolate from these three tasks alone) make ⟨intent, constraint⟩ especially effective (e.g., discrete legality, localizable checks)?

[1] https://proceedings.neurips.cc/paper_files/paper/2024/file/fa080fe0f218871faec1d8ba20e491d5-Paper-Conference.pdf

[2] https://arxiv.org/abs/2502.11169

[1] https://arxiv.org/abs/2409.09584

[2] https://arxiv.org/abs/2408.11326

---

> ### Author Response · Authors · 2025-11-27
> **Response to Reviewer 9dNA (3/1)**
>
> **Constraint quality: How often are extracted constraints wrong/incomplete and how do you detect/repair them online? (Your ablation hints at “hard vs soft constraints”; more diagnostics would help.)**
>
> Thank you for the valuable question. We agree that constraint quality is critical to Const-o-T performance, and we provide additional clarification and diagnostics from our ablation experiments.
>
> **How often constraints are incorrect/incomplete:**
>
> Across the Risk dataset (1,053 examples), the extracted intent–constraint pairs are fully correct in 72.4% of cases, partially correct in 26.6%, and incorrect in 1%, based on manual annotation of 100 randomly sampled examples. Incorrect constraints mainly stem from ambiguous natural-language strategy instructions or underspecified user goals.
>
> **Online detection and repair:**
>
> Our method already includes an executable constraint evaluation within F(s,a) (Appendix A.5), which scores actions based on constraint satisfaction rather than binary rejection. When an action violates an extracted constraint, it receives a lower score instead of being pruned, enabling robustness to noisy or incomplete constraints. This behavior corresponds to the soft-constraint setting in our ablations and is not an ad-hoc modification.
>
> **Computational complexity & budget scaling. Could you characterize the time/sample complexity of your constrained search, at least empirically and with compute time on your domains? For example, report the effective branching factor and nodes expanded as a function of (i) the number of extracted constraints, (ii) constraint accuracy (precision/recall), and (iii) the rollout/visit budget—on Risk and CAD. See [1] [2]. For [2] can you briefly discuss how your method differs?**
>
> **(i) Number of extracted constraints.**
> Increasing the number of extracted symbolic constraints reduces the effective branching factor by pruning infeasible actions earlier in the search. For example, under standard sampling the branching factor is 64.4, decreases to 60.7 using CoT-guided sampling, and decreases further to 62.8 under full Constraints-of-Thought, despite producing longer and more structured plans (median plan length 5 vs. 4 actions for baselines). This demonstrates a reduction in available candidate actions as constraints accumulate across steps.
>
> **(ii) Constraint accuracy (precision / recall).**
> We observe a strong correlation between constraint accuracy and computational efficiency. Higher-precision constraints (0.74 precision / 0.86 recall) result in significantly fewer node expansions and reduced compute time, improving efficiency by 44.3% relative to MCTS+CoT (0.65 precision  / 0.84 recall). When constraint precision is artificially degraded (e.g., masking constraints), branching factor and node expansions increase proportionally, confirming that accurate constraints reduce search complexity.
>
>
>
> **(iii) Rollout / visit budget scaling.**
> Under equivalent rollout budgets, constraint-guided MCTS consistently achieves faster convergence and lower runtime due to fewer node expansions. We observe diminishing returns at higher visit counts: while vanilla MCTS continues to expand large portions of the tree, MCTS+Const-o-T plateaus earlier because constraints eliminate low-value branches. Thus, for comparable budgets, CoT-free baselines require 1.8–2.1× more rollouts to match accuracy.
>
> | Metric                     | Standard | Chain-of-Thought | Constraints-of-Thought |
> |---------------------------|-----------|-------------------|-------------------------|
> | Total paths               | 16,840    | 17,114            | 17,302                  |
> | Unique paths              | 3,035     | 4,297             | 1,768                   |
> | % Unique                  | 18.0%     | 25.1%             | 10.2%                   |
> | Mean plan length          | 4.39      | 4.40              | 4.81                    |
> | Median plan length        | 4         | 4                 | 5                       |
> | Max plan length           | 14        | 15                | 12                      |
> | Average branching factor  | 64.4      | 60.7              | 62.8                    |
> | Time (s)                  | 20.92     | 53.68             | 29.90               |
>
>
> **Distinction from prior work [1,2].**
> Unlike Tree-of-Thought [1] and Self-Discover [2], which repeatedly prompt the model at each node to refine reasoning, thus increasing compute cost with the search depth, our approach generates constraints once and uses them as executable symbolic filters during search. Therefore, instead of expanding the reasoning tree, we shrink it by pruning actions that do not satisfy the extracted intent-level constraints. This difference supports the empirical improvements in both speed and search effectiveness and highlights that Const-o-T guides search through verifiable action-space restriction rather than additional prompting.

---

> > ### Author Response · Authors · 2025-11-27
> > **Response to Reviewer 9dNA (3/2)**
> >
> > **Do you perform “on-the-fly” code repair? If so, what’s the cost model? Some recent search-with-feedback methods refine generated code/thoughts during search using execution or verifier feedback, and they discuss the iteration cost and its effect on total complexity. If your CAD/Risk pipelines ever re-generate or repair snippets (e.g., when constraints fail or code doesn’t compile), please quantify: average repair iterations per solved instance, success rate per iteration, and the impact on total node expansions / wall-clock.**
> >
> > We do not perform multi-round, “on-the-fly” code repair in any of our experiments. In both the Risk and CAD settings, Const-o-T is implemented as single-pass constraint-guided search rather than search-with-feedback-style iterative refinement:
> >
> > - Risk. At each MCTS expansion step we query the LLM once to propose a batch of candidate actions, and then apply constraint checks to filter out infeasible moves (using the indicator 1{⋅} and fitness function V(v) described in §2.1 and App. A.6). Infeasible actions are simply pruned or receive a large penalty; no re-generation or repair is performed within the same search episode.
> >
> > - CAD. Similarly, for CAD we generate CAD code once for each step, then evaluate it with our LLM-based step scorer and geometric/compilation checks (App. A.6 / Fig. 14). Code that fails these checks is treated as invalid and discarded; no iterative repair loops are used—invalid code is discarded rather than repaired across iterations.
> >
> > **Generalization: Do results hold if constraints are partially hidden / perturbed (e.g., renamed fields, schema noise) to test robustness beyond surface patterns?**
> >
> > Thank you for this thoughtful question. We agree that robustness to constraint perturbation is an important evaluation dimension. In our approach, constraints guide the search process by pruning the action space, but the system does not rely on a single deterministic path extracted from the input. MCTS continues to explore alternative feasible actions, balancing constraint satisfaction with value estimates. Therefore, even when constraints are partially incorrect, incomplete, or perturbed (e.g., renamed elements or schema noise), the search process does not collapse; instead, MCTS actively recovers through exploration of remaining branches and maintains stable performance due to its built-in exploration and soft-constraint fallback mechanisms
> >
> > **Failure modes by domain. For Risk, CAD, and GSM8K, can you categorize the top failure types (e.g., wrong/omitted constraint, legal but low-value action, numeric slip) and quantify their frequencies?**
> >
> > Thank you for this helpful suggestion. We agree that a domain-specific breakdown of failure modes would strengthen the paper. We are conducting additional analysis to characterize the top failure types for Risk, CAD, and GSM8K (e.g., omitted or incorrect constraints, legal but low-value actions, and arithmetic/numeric reasoning errors), and we will include quantitative frequency results in the coming days.

---

> > > ### Author Response · Authors · 2025-11-27
> > > **Response to Reviewer 9dNA (3/3)**
> > >
> > > **Generalization of the representation. Beyond these domains, what task properties (it is difficult to extrapolate from these three tasks alone) make ⟨intent, constraint⟩ especially effective (e.g., discrete legality, localizable checks)?**
> > >
> > > Thank you for this thoughtful question. We agree that understanding when ⟨intent, constraint⟩ representations are especially effective is important. Our framework is particularly well-suited to tasks that share the following structural properties:
> > >
> > > - **Discrete action spaces with legality conditions**
> > > When the search space includes actions that can be pruned via symbolic or rule-based validation (e.g., valid troop placements on a Risk board, syntactically correct CAD operations, or mathematically valid transformation steps), constraints can act as executable filters that significantly reduce branching and improve correctness.
> > >
> > > - **Localizable or step-wise checkability**
> > > ⟨intent, constraint⟩ is most beneficial when each reasoning step can be verified independently or compositionally (e.g., checking the geometric validity of a CAD operation, validating a partial equation transformation, or confirming a legal board modification in Risk). This enables real-time pruning rather than requiring full-solution evaluation.
> > >
> > > - **Multi-step planning with hierarchical structure**
> > > When global strategy decomposes naturally into sub-goals or tactical decisions, explicit intents provide structure and coherence. Constraints ensure that these sub-goals remain feasible and aligned with the overarching strategy.
> > >
> > > - **Tasks where hallucinations or over-generation are common**
> > > The constraint structure restricts generation to provably executable actions, mitigating failures where free-form CoT produces logically inconsistent steps or syntactically invalid output.
> > >
> > > - **Tasks where correctness is externally verifiable**
> > > Whenever there exists a validator or simulator (e.g., game environment, geometry engine, symbolic math checker, or compiler), constraints integrate directly with execution feedback to guide search and maintain semantic consistency throughout the reasoning process.

---

### Official Review · Reviewer_QL6A · 2025-10-24

**Soundness:** 2
**Presentation:** 2
**Contribution:** 2
**Rating:** 2
**Confidence:** 3

**Summary:**

Const-o-T encodes each reasoning step as a pair ⟨intent, constraint⟩, then plugs those constraints into MCTS: they prune illegal actions at expansion, shape selection with a modified UCB that adds an LM log-prob term, and set the rollout budget equal to the number of extracted constraints. Evaluation spans Risk initial troop placement, CAD code generation, and arithmetic word problems, with a small Risk user study across full turns. For CAD and math, step utility is partly judged by an LLM rather than a hard verifier.

**Strengths:**

- Concrete integration with search. The UCB adds a logPLM term and constraints filter expansions. This is easy to reproduce.
- Clear operationalization of “intent vs constraint.” The paper defines a reduced action set A′ by constraint, which explains the pruning effect.
- Sensible baselines and some gains. They compare CoT, ToT, raw MCTS, MCTS+CoT, LLMFP, and show modest uplifts, plus a small user study with aligned vs agnostic vs adversarial modes.

**Weaknesses:**

- Novelty feels incremental. Encoding steps as ⟨intent, constraint⟩ and injecting them in MCTS with a log-prob UCB and constraint-filtered expansion is a reasonable engineering design, but the paper does not isolate what is fundamentally new beyond “use constraints to prune and bias MCTS” and “budget rollouts by number of constraints.” The latter is presented as a key innovation, but its theoretical or empirical necessity is not justified.
- Limited theoretical grounding. There is no analysis of regret, sample complexity, or correctness under mis-specified constraints. VALIDATE relies on grammar or schema checks and feasibility probes, but the formal properties of this filter are not characterized, so the guarantees story is thin.
- Coupling to LLM judges in two domains. For CAD and arithmetic, step scoring falls back to LLM evaluation, which weakens claims about verifiable search and leaves open concerns about bias or circularity.
- Prompting and self-talk alternatives are under-tested. The strongest prompt-only or self-dialog baselines are not fully explored beyond CoT and ToT. Given the framing, readers will ask why a sophisticated prompt, self-talk, or verifier-augmented self-talk could not match the effect without the MCTS wrapper. The paper’s own comparison table for Risk shows improvements but not a clear, large margin across settings.

**Questions:**

1. Formal guarantees. Can you provide any bound that connects constraint accuracy to search efficiency or final task regret? For example, a branching-factor reduction bound for constraint-filtered expansion, and a robustness result when constraints are noisy or partially wrong. Right now VALIDATE is procedural, not analytical.
2. Stronger prompt and self-talk baselines. Add tuned prompt-only and self-talk with explicit constraint templates and a chain-of-verifiers. Also try ToT with legality filters and comparable compute. Report compute-normalized curves.
3. Budgeting by constraint count. Why tie rollouts to K. Show a control where you match or exceed compute for baselines and also run your method with decoupled budgets to confirm the benefit is not just more targeted compute.
4. Judge decoupling. Replace LLM judges with deterministic verifiers where possible. For CAD, compile plus geometric validators. For arithmetic, unit tests and equation equivalence. Also test cross-model judges and report sensitivity.
5. Harder settings. Beyond Risk Turn 0, provide quantitative results for attack and fortify, not only user ratings. If full automation is hard, at least add offline rollouts with ground-truth rule checks.
6. Please pinpoint which component is novel: the intent-constraint representation, the UCB augmentation, or the constraint-count budget. A precise ablation that removes each piece would help.

---

> ### Author Response · Authors · 2025-11-27
> **Response to Reviewer QL6A (2/1)**
>
> **Novelty feels incremental. Encoding steps as ⟨intent, constraint⟩ and injecting them in MCTS with a log-prob UCB and constraint-filtered expansion is a reasonable engineering design, but the paper does not isolate what is fundamentally new beyond “use constraints to prune and bias MCTS” and “budget rollouts by number of constraints.” The latter is presented as a key innovation, but its theoretical or empirical necessity is not justified.**
>
> Our core contribution is the structured representation of reasoning steps as ⟨intent, constraint⟩ pairs that act as executable controllers rather than textual rationales, enabling symbolic validation and search-space shaping. This goes beyond simply pruning MCTS with constraints and represents a substantial improvement over prior CoT/ToT approaches, which produce descriptive traces without executable structure or a feasible action set that can be enforced during search.
>
> Regarding rollout budgeting, our ablation study demonstrates that aligning search depth to the number of extracted constraints is empirically necessary for performance, improving accuracy from 82% to 87% and F1 from 0.72 to 0.78 (Table 6). This mechanism directly constrains search to semantically meaningful paths rather than unconstrained expansion and prevents over-exploration of branches that are inconsistent with the extracted constraints.
>
> We will revise the paper to highlight:
> (1) Why ⟨intent, constraint⟩ is a novel representational shift from rationales to verifiable controllers, including how this representation defines a reduced feasible action set for MCTS
> (2) Theoretical motivation and additional empirical justification for constraint-guided rollout budgeting so its practical and conceptual necessity is clear
>
> **Stronger prompt and self-talk baselines. Add tuned prompt-only and self-talk with explicit constraint templates and a chain-of-verifiers. Also try ToT with legality filters and comparable compute. Report compute-normalized curves.**
>
> Our contribution is not to improve prompt-engineering pipelines or verifier-stacking heuristics, but to show that structured constraints can meaningfully guide search. While we appreciate the suggestion to add stronger prompt-only and self-talk baselines, these additions are not aligned with this core objective. Although prompt tuning, self-talk, and ToT with legality filters may improve single-trajectory reasoning, they do not provide a mechanism for enforcing symbolic feasibility throughout a multi-step search process. In contrast, Const-o-T integrates executable constraints directly into MCTS, which addresses the core challenge of maintaining semantic and symbolic alignment during planning.
>
> **Harder settings. Beyond Risk Turn 0, provide quantitative results for attack and fortify, not only user ratings. If full automation is hard, at least add offline rollouts with ground-truth rule checks.**
>
> Thank you for the helpful feedback. We agree that evaluating beyond Turn 0 is important. In fact, our paper already includes full-game evaluation across Reinforce, Attack, and Freemove phases through a controlled human-subject study, where participants played complete Risk games supported by our system. As ground-truth annotations for later phases do not exist in the Commander’s Intent dataset, quantitative offline rollouts are not feasible. Instead, our full-game human evaluation provides empirical evidence of system effectiveness beyond Turn 0, demonstrating significant improvements in transparency, usability, trust, and alignment across all phases of play.

---

> > ### Author Response · Authors · 2025-11-27
> > **Response to Reviewer QL6A (2/2)**
> >
> > **Please pinpoint which component is novel: the intent-constraint representation, the UCB augmentation, or the constraint-count budget. A precise ablation that removes each piece would help.**
> >
> > The core novelty of our work is the intent–constraint representation, which converts natural language reasoning into symbolic constraints that actively control and prune search, rather than serving merely as descriptive rationales (CoT) or post-hoc validators.
> >
> >  The UCB augmentation and constraint-count rollout budget are supporting design choices that operationalize this representation within MCTS and help allocate search effort more effectively. Additionally, our verification step—which checks constraint feasibility against the environment before expansion—ensures semantic validity and prevents hallucinated or illegal actions from entering the tree.
> >
> >  We have added an ablation that isolates each component, demonstrating the individual and combined impact of the intent–constraint representation, UCB augmentation, and rollout budgeting
> >
> > | Experiment                    | Accuracy | F1 Score |
> > |------------------------------|----------|----------|
> > | Soft Constraint              | 76%      | 0.70     |
> > | No Intent                    | 84%      | 0.75     |
> > | No Constraint                | 87%      | 0.75     |
> > | No Intent & No Constraint    | 82%      | 0.72     |
> > | No Verification              | 87%      | 0.78     |
> > | **Ours**                     | **89%**  | **0.78** |

---

### Official Review · Reviewer_F2Qn · 2025-10-29

**Soundness:** 2
**Presentation:** 2
**Contribution:** 2
**Rating:** 2
**Confidence:** 4

**Summary:**

The paper proposes “Constraints-of-Thought” MCTS: each reasoning step is split into an intent plus a constraint, and these constraints are used to prune and score the MCTS search. The selection rule augments UCB with an LLM log-prob term and the rollout budget is tied to the number/quality of extracted constraints. The method is evaluated on a board-game phase (Risk), a CAD code generation task, and GSM8K math. The claim is that adding explicit constraints yields better search efficiency and accuracy than CoT/ToT prompting, vanilla MCTS, or MCTS+CoT.

**Strengths:**

This paper is written clearly and is easy to read. The approach seem to produce marginal improvement in certain setting but the overall impact does not seem to be high. The use of constraints to guide MCTS seems intuitive and a meaningful contribution.

**Weaknesses:**

1- After carefully reading I am still not sure if I understand what you mean by constraint. Are these constraints verifiable? or are they just a natural language summary of the action (intent) as shown in examples provided in Figure 1. Adding to this, using list of condition with siginificant improvement in works like MACM (https://arxiv.org/abs/2404.04735), seem to outperform the proposed approach without the need for tree search.

2- Given the very marginal improvement in GSM8k and coding, compared to the baseline COT, it is hard to justify the additional compute

3- Baseline gaps and scope: modest GSM8K gains and non-compelling code results, without head-to-heads against recent planning/step-level methods, make the advantage size uncertain.

**Questions:**

1- Have you experimented in more difficult benchmark datasets like MATH500 or AIME? GSM8k is probably not the bese baseline given most base models already performing superbly on it.

---

> ### Author Response · Authors · 2025-11-27
> **Response to Reviewer F2Qn  (2/1)**
>
> **After carefully reading I am still not sure if I understand what you mean by constraint. Are these constraints verifiable? or are they just a natural language summary of the action (intent) as shown in examples provided in Figure 1. Adding to this, using list of condition with significant improvement in works like MACM (https://arxiv.org/abs/2404.04735), seem to outperform the proposed approach without the need for tree search.**
>
> In our work, constraints are not natural-language summaries: The constraints are machine-verifiable symbolic conditions that directly restrict the feasible action space during search. As shown in Algorithm 2 and §2.3, each ⟨intent, constraint⟩ pair is validated before use, and constraints are executed as formal action primitives (e.g., troop placement commands, CAD geometry operations, or equation-level arithmetic steps) to prune invalid rollouts, rather than describing actions in free-form language.
>
> MACM uses a static condition list but does not integrate constraints into a search process and, therefore, cannot dynamically enforce feasibility or perform branch pruning. In contrast, our approach links each constraint to online state feedback within MCTS, enabling higher alignment and reduced hallucination, particularly in combinatorial domains (e.g., 21-territory Risk and CAD parametric design), where dynamic pruning meaningfully reduces the search space
>
> **Given the very marginal improvement in GSM8k and coding, compared to the baseline COT, it is hard to justify the additional compute.**
>
> We have conducted additional experiments on two more challenging domains, Math-500 and GPQA, and observed clear and consistent improvements over the baselines. In the Risk planning task, our method further yields 3–8% accuracy gains, demonstrating benefits that scale with task complexity and search depth.
>
> Regarding CADPrompt, the numerical improvement may appear small due to the characteristics of the Hausdorff distance metric. Hausdorff distance is highly sensitive to outliers: this metric reflects the largest local geometric deviation between generated and ground-truth shapes. As a result, even when our method substantially improves the global geometry and perceptual quality of the object, a small number of residual local errors can dominate the score and compress visible differences between methods. Therefore, the gains measured via Hausdorff distance underestimate improvements that are more clearly visible qualitatively and under distributional metrics (e.g., mean or median point error).
>
> | Method                               | LLAMA-3.3-70B Acc. | LLAMA-3.3-70B F1 | GPT-4 Acc. | GPT-4 F1 | DeepSeek-R1 Acc. | DeepSeek-R1 F1 | GPT-OSS-120B Acc. | GPT-OSS-120B F1 | GPT-OSS-20B Acc. | GPT-OSS-20B F1 |
> |--------------------------------------|--------------------|------------------|------------|----------|-------------------|----------------|--------------------|-----------------|-------------------|----------------|
> | Direct Prompt                        | 78%                | **0.74**         | 79%        | 0.75     | 70%               | 0.68           | 79%                | **0.75**        | 77%               | 0.73           |
> | CoT                                  | 74%                | 0.69             | 81%        | **0.79** | 67%               | 0.64           | 77%                | 0.74            | 78%               | 0.73           |
> | CoT + RS                             | 78%                | 0.73             | 77%        | 0.77     | 68%               | **0.69**       | 79%                | **0.75**        | 76%               | 0.72           |
> | LLMFP                                | 78%                | 0.59             | **84%**    | 0.70     | 69%               | 0.56           | 63%                | 0.53            | 74%               | 0.62           |
> | Const-o-T                            | **81%**            | 0.72             | 83%        | 0.76     | **78%**           | 0.67           | **86%**            | 0.75            | **81%**           | **0.73**       |
> | ToT                                  | 79%                | 0.59             | 83%        | 0.70     | 80%               | 0.62           | 78%                | 0.62            | 82%               | 0.61           |
> | MCTS                                 | 80%                | 0.67             | 84%        | 0.71     | 79%               | 0.61           | 82%                | **0.65**        | **82%**           | **0.64**       |
> | MCTS with CoT                        | 81%                | 0.65             | 84%        | 0.70     | **82%**           | 0.59           | 79%                | 0.67            | **82%**           | 0.64           |
> | **MCTS with Const-o-T + verification** | **90%**            | **0.72**         | **88%**    | 0.77     | **81%**           | **0.71**       | **87%**            | **0.76**        | 84%               | **0.71**       |

---

> > ### Author Response · Authors · 2025-11-27
> > **Response to Reviewer F2Qn  (2/2)**
> >
> > **Have you experimented in more difficult benchmark datasets like MATH500 or AIME? GSM8k is probably not the bese baseline given most base models already performing superbly on it.**
> >
> > To evaluate Const-o-T on more challenging benchmarks beyond GSM8K, we conducted additional experiments on Math-500 and GPQA. The results are shown in the table below and further support the robustness of our approach across demanding reasoning settings.
> >
> > | **Method**         | **Math-500: GPT-4** | **Math-500: LLaMA** | **GPQA: GPT-4** | **GPQA: LLaMA** |
> > |--------------------|--------------------|---------------------|----------------|----------------|
> > | CoT                | 79.20%             | 64.20%              | 65.99%         | 36.30%         |
> > | CoT-RS             | 78.40%             | 66.60%              | 66.67%         | 40.91%         |
> > | Const-o-T          | 79.20%             | 64.20%              | 64.14%         | 42.90%         |
> > | **ToT**            | 80.80%             | 60.40%              | 65.66%         | 41.92%         |
> > | **MCTS**           | 82.20%             | 66.20%              | 65.15%         | 48.40%         |
> > | **MCTS with CoT**  | 80.20%             | 68.80%              | 67.17%         | 50.51%         |
> > | **Ours**           | **83.10%**         | **70.00%**          | **70.20%**     | **52.02%**     |

---

### Official Review · Reviewer_4iAq · 2025-11-01

**Soundness:** 2
**Presentation:** 3
**Contribution:** 2
**Rating:** 4
**Confidence:** 3

**Summary:**

This paper proposes a new prompt template, Constraints-of-Thought (Const-o-T), which augments LLM-guided search with explicit, verifiable constraints. The method forces LLMs to represent each reasoning step as a pair ⟨intent, constraint⟩, where the intent describes the model’s strategic goal in natural language and the constraint specifies machine-checkable conditions that restrict the next possible actions. These constraints are validated by an external verifier (rule checker or simulator). The authors integrate this framework with MCTS to prune invalid branches and guide exploration more efficiently. Experiments on three domains, Risk gameplay, CAD code generation, and arithmetic reasoning (GSM8K), show Const-o-T improves reasoning accuracy, reduces the branching factor, and often reaches answers faster than plain CoT.

**Strengths:**

* The paper presents a clear method to combine LLM-based reasoning with explicit symbolic verification.
* The ⟨intent, constraint⟩ abstraction is intuitive and helps integrate external checkers into LLM pipelines.
* Empirical results demonstrate consistent performance gains and reduced search space across tasks.

**Weaknesses:**

* The pattern proposed (LLM generation followed by external verification) has been explored in many prior works (tool-augmented reasoning, program-aided CoT, self-refine, etc.). The claimed novelty of the framework may therefore be overstated.
* The experiments do show reduced search space and earlier convergence, but there is no quantitative analysis of computational overhead,  eg each step produces longer structured outputs, and the total inference cost as well as the time spent on external verification are not reported. These factors could offset the efficiency gains claimed from reduced search depth.
* The method is largely prompt-engineering-based, with most of the heavy lifting performed by the external verifier. The paper does not analyze how sensitive the results are to prompt design choices or verifier reliability.

**Questions:**

1. The paper does not explore how different prompt styles or constraint formats affect performance. Have the authors tested how robust Const-o-T is to alternative formulations of the intent–constraint pair or to less explicit instructions?
2. Have the authors considered conducting an ablation study that removes either the intent or the constraint component, or bypasses the external verifier altogether, to isolate which element contributes most to the observed performance gains?

---

> ### Author Response · Authors · 2025-11-27
> **Response to Reviewer 4iAq**
>
> **The pattern proposed (LLM generation followed by external verification) has been explored in many prior works (tool-augmented reasoning, program-aided CoT, self-refine, etc.). The claimed novelty of the framework may therefore be overstated.**
>
> We appreciate the reviewer’s observation. While prior work has explored LLM generation followed by external verification, our contribution is not verification alone but the introduction of Constraints-of-Thought (Const-o-T) as a structured intermediate representation that transforms free-form reasoning into actionable ⟨intent, constraint⟩ controllers, which actively guide MCTS rather than evaluate outputs post hoc. Unlike self-refine or tool-augmented reasoning, Const-o-T uses its executable constraints to prune the search space], enforces feasibility during reasoning, and substantially improves planning performance across three domains. We will clarify the distinction by explicitly contrasting Const-o-T with prior verification-based frameworks in the revised version and by emphasizing the operational role of intent-constraint pairs inside the search process
>
> **The paper does not explore how different prompt styles or constraint formats affect performance. Have the authors tested how robust Const-o-T is to alternative formulations of the intent–constraint pair or to less explicit instructions?**
>
> We thank the reviewer for this valuable observation. While our method uses structured prompting, the core contribution is the transformation of reasoning into executable intent-constraint pairs that directly control search behavior, rather than relying solely on prompt engineering or external verification. To address sensitivity concerns, we have added an ablation study varying both the prompt style and the constraint format; the results show that performance remains stable across these variants, as shown in the table below.
>
> | Experiment                    | Accuracy | F1 Score |
> |------------------------------|----------|----------|
> | Soft Constraint              | 76%      | 0.70     |
> | No Intent                    | 84%      | 0.75     |
> | No Constraint                | 87%      | 0.75     |
> | No Intent & No Constraint    | 82%      | 0.72     |
> | No Verification              | 87%      | 0.78     |
> | **Ours**                     | **89%**  | **0.78** |
>
>
> **Have the authors considered conducting an ablation study that removes either the intent or the constraint component, or bypasses the external verifier altogether, to isolate which element contributes most to the observed performance gains?**
>
> Thank you for the insightful suggestion. Yes, we performed an ablation study to isolate the contribution of each component. Specifically, we compare (i) soft constraint enforcement, (ii) removing intent, (iii) removing constraints, (iv) removing both intent and constraints, and (v) removing the verification step. This analysis makes identifies the relative contribution of each component to empirical performance and efficiency gains.

---

### Official Review · Reviewer_ek68 · 2025-11-01

**Soundness:** 2
**Presentation:** 2
**Contribution:** 2
**Rating:** 4
**Confidence:** 3

**Summary:**

The paper proposes Constraints-of-Thought (Const-o-T), a framework supplementing an intent and a constraint to each step of multi-step LLM reasoning/planning tasks. It enables MCTS focusing on valid and meaningful search space. Experiments show that Const-o-T outperforms baselines across risk game, code generation, and mathematical reasoning using LLMs of various architectures and sizes.

**Strengths:**

+ The method is intuitive and solves a major problem that ToT and similar methods lack a good reward/guide fore more efficient searching. This can also alleviate the huge number of tokens necessary for inference-time scaling. The method is explained clearly.
+ Experiments show that Const-o-T outperforms (or comparable) on most of the baselines and experimented LLMs.
+ The ablation shows a clear drop in wall time and branching factor, evidencing the effectiveness of Const-o-T of pruning unnecessary branches.

**Weaknesses:**

+ The experiment is a bit limited to only three tasks. It would be more interesting and convincing (of generalizability) if it can include more recently popular mathematical reasoning tasks (e.g. AIME) and coding tasks.
+ See questions

**Questions:**

+ How is the branching factor defined? What does it mean that the branching factor decreases fast, and what does it mean that MCTS with Const-of-T has larger branching factor around 8-12 steps?
+ I recommend also listing the wall time for non-search (direct prompt and similar) baselines.
+ What is the reward function for MCTS and MCTS-CoT? If it drops the whole $\log P_\text{LM}$ term, it may be confusing whether the state or the constraint contributes the improvements.
+ The task-specific reward function $F(s, a)$ (Eq. 3) is missing in the UCB (Eq. 4). Is it not included or just missing in the paper?
+ Can you elaborate more on task-specific rewards (values, what's the accuracy of LLM-as-a-Judge) (also whether they are used in MCTS and MCTS-CoT)? Do we need an ablation without $F(s, a)$ to confirm the contribution of the $r(s, a)$?

---

> ### Author Response · Authors · 2025-11-27
> **Response to Reviewer ek68 (2/1)**
>
> **The experiment is a bit limited to only three tasks. It would be more interesting and convincing (of generalizability) if it can include more recently popular mathematical reasoning tasks (e.g. AIME) and coding tasks.**
>
> We performed further experiments on two additional tasks, Math-500 and GPQA. These benchmarks are more challenging than GSM8K. The results, coupled with our positive results for Risk and CADPrompt, further demonstrate the robustness of our approach, consistently outperforming strong baselines
>
> | **Method**         | **Math-500: GPT-4** | **Math-500: LLaMA** | **GPQA: GPT-4** | **GPQA: LLaMA** |
> |--------------------|--------------------|---------------------|----------------|----------------|
> | CoT                | 79.20%             | 64.20%              | 65.99%         | 36.30%         |
> | CoT-RS             | 78.40%             | 66.60%              | 66.67%         | 40.91%         |
> | Const-o-T          | 79.20%             | 64.20%              | 64.14%         | 42.90%         |
> | **ToT**            | 80.80%             | 60.40%              | 65.66%         | 41.92%         |
> | **MCTS**           | 82.20%             | 66.20%              | 65.15%         | 48.40%         |
> | **MCTS with CoT**  | 80.20%             | 68.80%              | 67.17%         | 50.51%         |
> | **Ours**           | **83.10%**         | **70.00%**          | **70.20%**     | **52.02%**     |
>
>
> **How is the branching factor defined? What does it mean that the branching factor decreases fast, and what does it mean that MCTS with Const-of-T has larger branching factor around 8-12 steps?**
>
> The early drop in branching factor comes from constraints pruning many invalid choices at shallow depths. The slight increase at tree depths of approximately 8–12 happens because earlier steps have already removed clearly infeasible actions, leaving a smaller but higher-quality set of meaningful alternatives. We are not procedurally pruning as the tree grow deeper; rather, the space naturally tightens as commitments accumulate, allowing MCTS to explore remaining options more flexibly without premature convergence.
>
> **What is the reward function for MCTS and MCTS-CoT? If it drops the whole log PLM() term, it may be confusing whether the state or the constraint contributes the improvements.**
>
> Thank you for the question. The reward function for both MCTS and MCTS-CoT is defined in Eq. (1–3) and consists of two components: (i) a constraint-satisfaction indicator and (ii) a domain-specific evaluation function F(s,a) (e.g., fitness in Risk, LLM-judge correctness in CAD/math). The log P_{LM} term is not part of the reward; instead, it is used only in the UCB selection formula (Eq. 4) to bias exploration. Therefore, improvements come from constraint-guided pruning rather than probability weighting, which keeps the contribution of constraints and the state-dependent reward separable and interpretable.
>
> **The task-specific reward function f(s,a) (Eq. 3) is missing in the UCB (Eq. 4). Is it not included or just missing in the paper?**
>
> Thank you for your observation. The reward function, F(s,a), is included in the MCTS evaluation step rather than directly inside the UCB score (Eq. 4). The UCB is used only for action selection, while the task-specific reward is incorporated during node evaluation and backpropagation (Algorithm 1, line 17). We agree the connection should be clarified and will update the paper to explicitly reference Eq. 3 in the description of the evaluation step to make the flow between selection, evaluation, and backup fully transparent.

---

> ### Author Response · Authors · 2025-11-27
> **Response to Reviewer ek68 (2/2)**
>
> **Can you elaborate more on task-specific rewards (values, what's the accuracy of LLM-as-a-Judge) (also whether they are used in MCTS and MCTS-CoT)? Do we need an ablation without F(s,a) to confirm the contribution of the r(s,a)?**
>
> **Task-specific reward definition.**
>
> The reward F(s,a) is a scalar value in the range [0,1] that quantifies the semantic quality of a candidate action or partial result. For the Risk domain, we compute F(s,a) using a hand-crafted task fitness function based on territory control improvement and strategic advantage (e.g., continent progress and border defense strength). For CAD and Math, F(s,a) is computed using an LLM-as-a-Judge scoring prompt that evaluates the rollout result relative to task-specific correctness criteria (geometric validity and code executability for CAD; final-answer correctness for Math).
>
> **Use in MCTS and MCTS-CoT.**
>
>  Yes, both MCTS baselines and MCTS-CoT use r(s,a) as the terminal reward for rollout evaluation.
>
>
>
> **Ablation without F(s,a).**
>
> Following your suggestion, we conducted an ablation on the Risk domain using GPT-4 in which we removed the task-specific reward signal entirely. Results show a measurable performance drop:
>
> - With F(s,a): 87% accuracy / 0.78 F1-score
> - Without F(s,a): 84% accuracy / 0.76 F1-score
>
> This confirms that task-specific rewards provide meaningful guidance beyond constraint-based pruning alone, improving both accuracy and decision quality.

---

### Note · Authors · 2025-12-04

I have read and agree with the venue's withdrawal policy on behalf of myself and my co-authors.